

# The Chalmers Cloud Ice Climatology: Retrieval implementation and validation

Adrià Amell[1, *], Simon Pfreundschuh[1, 2, *], and Patrick Eriksson[1]

[1]Department of Space, Earth and Environment, Chalmers University of Technology, Gothenburg, Sweden
[2]Department of Atmospheric Science, Colorado State University, Fort Collins, USA
[*]These authors contributed equally to this work.

**Correspondence:** Simon Pfreundschuh (simon.pfreundschuh@colostate.edu)

**Abstract.**

Ice clouds are a crucial component of the Earth's weather system, and their representation remains a principal challenge for current weather and climate models. Several past and future satellite missions were explicitly designed to provide observations offering new insights into cloud processes, but these specialized cloud sensors are limited in their spatial and temporal coverage.

Geostationary satellites have been observing clouds for several decades and can ideally complement the sparse measurements from specialized cloud sensors. However, the geostationary observations that are continuously and globally available over the full observation record are restricted to a small number of wavelengths, which limits the information they can provide on clouds.

The Chalmers Cloud Ice Climatology (CCIC) addresses this challenge by applying novel machine-learning techniques to
retrieve ice cloud properties from globally gridded, single-channel geostationary observations that are readily available from 1980 onwards. CCIC aims to offer a novel perspective on the record of geostationary IR observations by providing spatially and temporally continuous retrievals of the vertically-integrated and vertically-resolved concentrations of frozen hydrometeors, typically referred to as ice water path (IWP) and ice water content (IWC). In addition to that, CCIC provides 2D and 3D cloud masks and a 3D cloud classification.

A fully convolutional quantile regression neural network constitutes the core of the CCIC retrieval, providing probabilistic estimates of IWP and IWC. The network is trained against CloudSat retrievals using 3.5 years of global collocations. Assessment of the retrieval accuracy on a held-out test set demonstrates considerable skill in reproducing the reference IWP and IWC estimates. In addition, CCIC is extensively validated against both in-situ and remote sensing measurements from two flight campaigns and a ground-based radar. The results of this independent validation confirm the ability of CCIC to retrieve
IWP and, to first order, even IWC. CCIC thus ideally complements temporally and spatially more limited measurements from dedicated cloud sensors by providing spatially and temporally continuous estimates of ice cloud properties. The CCIC network and its associated software are made accessible to the scientific community.



## 1 Introduction

The representation of clouds and convection in weather and climate models is recognized as a critical factor limiting the
accuracy of forecasts of future weather and climate (Bony et al., 2015; Brunet et al., 2010). More specifically, the latest decadal
survey of the National Academy of Science (National Academies of Sciences, Engineering, and Medicine, 2018) identified the
question '*Why do convective storms, heavy precipitation, and clouds occur exactly when and where they do?*' as a key science
question for the upcoming decade.

Several upcoming satellite missions and sensors will provide measurements of various cloud properties in order to provide
new observations to improve the understanding of cloud processes and their representation in numerical models. The EarthCare
mission (Illingworth et al., 2015) will continue the combined radar-lidar observations from the CloudSat and Calipso and be
the first space-borne sensor to measure vertical velocities of hydrometeors. In a similar vein, the Investigation of Convective
Updrafts (INCUS) and NASA's Atmosphere Observation System (AOS) will provide measurements of the evolution convective
storms by means of a constellation space-borne radars. Furthermore, sub-millimeter wave observations from the upcoming Ice
Cloud Imager (ICI) on board the next generation of European polar-orbiting operational weather satellites will improve space-
borne measurements of ice hydrometeor concentrations (Eriksson et al., 2020) by providing observations at sub-millimeter
wavelengths.

While these and other future satellite missions will provide crucial, novel observations to further the understanding of con-
vection and clouds, it will take several years before their observations will be available and the observational record sufficiently
extensive to derive meaningful results. For the study of processes on annual and decadal scales it is therefore necessary to find
ways to make better use of observations with a long record of availability. Furthermore, many satellite missions are limited
in their spatial and/or temporal sampling. EarthCare and ICI, for example, will both be in sun-synchronous orbits and their
observations will therefore not be capable of resolving the diurnal cycle of cloud processes.

In principle, observations from geostationary satellites would be ideal to complement sparse measurements from specialized
cloud sensors. However, only a limited number of channels are consistently globally available. Moreover, these observations
are limited to the visible and infrared wavelengths. Since the information in these channels stems mostly from the top of clouds,
it is generally understood that these observations are insufficient to meaningfully constrain the distribution of hydrometeors
throughout the full atmosphere. However, Amell et al. (2022) showed that reasonably accurate estimates of the vertically-
integrated concentration of ice hydrometeors, the *total ice water path* (TIWP), can be obtained from geostationary IR channels
and retrievals based on modern deep learning techniques.

TIWP shall be understood here as the vertically-integrated amount of all types of frozen hydrometeors. In currently avail-
able ice water path retrievals based on passive observations it is not always clearly defined which type of hydrometeors are
considered. As a consequence, TIWP is not very well constrained by currently available observations and there remain large
differences in ice hydrometeor concentrations between different models (Waliser et al., 2009; Eliasson et al., 2011; Duncan and
Eriksson, 2018). Estimates of TIWP differ widely between currently available observational datasets (Duncan and Eriksson,
2018). While this is, at least partly, due to the inherent limitations of different observing techniques and the significant impact



of uncertain microphysical assumptions on TIWP estimates, additional factors limit the potential of currently available datasets to inform studies of cloud processes. For example, datasets derived from sensors in sun-synchronous orbits such as the Cloud-Sat CPR, MODIS or AVHRR are typically not able to resolve the diurnal cycle of clouds. On the other hand, datasets derived from passive microwave sensors, only capture precipitating ice particles and were found to be at low end of the spectrum of global TIWP estimates. Although being derived from geostationary sensors, and thus capable of resolving the diurnal cloud cycle, TIWP estimates from the ISCCP dataset were found to be very low and not agree well with spatial distribution inferred from CloudSat measurements (Eliasson et al., 2011).

CCIC aims to revisit the historical record of geostationary infrared (IR) observations to investigate the potential of these observations to provide improved constraints on the concentration of ice hydrometeors in the atmosphere. CCIC is motivated by the findings from Pfreundschuh et al. (2022c) and Amell et al. (2022), which showed that neural-network-based retrievals that leverage the spatial structure of satellite observations can achieve considerable higher accuracy than traditional methods. CCIC retrieves TIWP and several other cloud properties from a single IR window channel with a wavelength of around $11\,\mu m$. Although these observations primarily provide information on the temperature of the atmosphere at the cloud top, gridded datasets of observations at $11\,\mu m$ are readily available from 1980 onwards (Knapp et al., 2011), which significantly simplifies producing a long time series of spatially and temporally continuous TIWP measurements. The primary aim of CCIC is to provide an updated, comprehensive, and easily accessible record of TIWP estimates that can provide context for more accurate but generally more sparse measurements from upcoming satellite missions targeting cloud properties.

This article presents the underlying retrieval algorithm of CCIC and validates it against independent measurements of cloud concentrations. Section 2 describes the dataset that was created for the training of the deep neural network used by CCIC and introduces the reference measurements used to validate the retrieval. Section 3 establishes the nominal accuracy of the retrieval by evaluating it on a held-out test dataset, while Sect. 4 assesses the retrieval against independent in-situ and remote sensing measurements of hydrometeor concentrations. Finally, Sect. 5 discusses the validation results and potential applications of CCIC and Sect. 6 summarizes the principal conclusions from this study.

## 2 Methods and Data

The CCIC retrieval is based on a convolutional neural network (CNN) that leverages quantile regression (Pfreundschuh et al., 2018) to provide probabilistic estimates of TIWP as well as additional cloud properties. The following sub-sections describe the implementation and training of the neural-network-based retrieval. Following this, the independent measurements used to validate the CCIC retrieval are presented.

### 2.1 The CCIC retrieval

In order to apply the retrieval to the extensive historical record of geostationary IR observations, CCIC was designed to use 11 µm IR brightness temperatures as the only retrieval input. The retrieval does not ingest any ancillary data to make the estimates independent from other datasets. As mentioned in the introduction, TIWP constitutes the primary retrieval target





**Table 1.** CCIC retrieval targets. CCIC provides probabilistic estimates of the cloud properties listed in this table. Due to storage limitations only the statistics listed under 'Retrieval output' are actually retained as output. The cloud classification is based on the nine cloud classes from the CloudSat 2B-CLDCLASS product (no cloud, cirrus (Ci), altostratus (As), altocumulus (Ac), stratus (St), stratocumulus (Sc), cumulus (Cu), nimbostratus (Ns), and deep convection (DC)).

| Target | Retrieved quantity | Retrieval output | Vertical levels |
|---|---|---|---|
| TIWP | $p(\text{TIWP} \mid T_{B,11\,\mu m})$ | Expected value, credible interval (CI) bounds | 1 |
| TIWC | $p(\text{TIWC} \mid T_{B,11\,\mu m}$ | Expected value | 20 |
| Cloud mask | $P(\text{Cloud anywhere in column} \mid T_{B,11\,\mu m})$ | $P(\text{Cloud anywhere in column} \mid T_{B,11\,\mu m})$ | 1 |
| Cloud class | $P(\text{Cloud class} \mid T_{B,11\,\mu m})$ | $P(\text{Cloud in parcel} \mid T_{B,11\,\mu m})$, Most likely cloud type | 20 |

**Table 2.** Spatiotemporal resolution and coverage of the input data products.

| | CPCIR | GridSat |
|---|---|---|
| Spatial resolution | $0.036°$ | $0.07°$ |
| Temporal resolution | 30 min | 180 min |
| Temporal coverage | 2000 – present | 1980 – present |
| Spatial coverage | $60°$ S – $60°$ N | $70°$ S – $70°$ N |

of CCIC. However, it is accompanied by the vertically-resolved concentration of ice hydrometeors, referred to as *total ice*
*water content* (TIWC). Analogously to TIWP, the name TIWC was chosen to emphasize that the estimates of total mass of frozen hydrometeors are not restricted to a single species. Additional, secondary retrieval targets provided by CCIC are a vertically-resolved and a vertically-integrated cloud masks. Table 1 summarizes the CCIC retrieval targets.

### 2.1.1 Input data

The CCIC retrieval ingests geostationary IR input data from two distinct datasets in order to maximize the temporal cover-
age and the temporal and spatial resolution of the CCIC data record. The first dataset is the GridSat-B1 product version 2 (GridSat, Knapp et al. (2011); Knapp and NOAA CDR Program (2014)), which covers the time from 1980 until the present at a temporal resolution of $3\,\text{h}$ and a spatial resolution of $0.07°$. The second dataset is the NOAA Climate Prediction Center globally merged IR product version 1 (CPCIR, Janowiak et al., 2001, 2017), which is available only from the year 2000 but offers higher temporal and spatial resolution of $30\,\text{min}$ and $0.036°$, respectively. Both datasets provide merged and gridded
IR brightness temperatures from the channels closest to $11\,\mu m$ from the global constellation of historical and current geostationary meteorological satellites. The datasets apply intersatellite normalization and, in addition, GridSat applies a temporal normalization targeting long historical analyses (Knapp et al., 2011). The IR radiances are used as input to the CCIC network without discerning between the datasets. The characteristics of the two datasets are summarized in Table 2.





### 2.1.2 Training data

The reference data for the CCIC retrieval targets is derived from two CloudSat products: the level 2 cloud scenario classification version R05 (2B-CLDCLASS, Sassen and Wang, 2008) and the level 2 CloudSat and CALIPSO ice cloud property version R05 (2C-ICE, Deng et al., 2010, 2013, 2015). The 2B-CLDCLASS and 2C-ICE granules were both collocated with the input data and used to extract the following information for each profile: TIWP, TIWC, a 2D cloud mask indicating the presence of a cloud anywhere in the vertical profile, and vertically resolved cloud classification following the 2B-CLDLASS product. The

cloud classification assigns one of nine possible classes to each CloudSat radar bin (Table 1). Although there is redundancy in these variables, as TIWC integrates to TIWP and the 2D cloud mask can, in principle, be derived from the vertically-resolved cloud classification, they were kept separately.

Despite the vertical information in the 2B-CLDCLASS and 2C-ICE products is provided in bins of $240\,\mathrm{m}$, the reference data was regridded to a uniform altitude grid with a vertical resolution of $1\,\mathrm{km}$ relative to the surface of the digital elevation model

provided in these products. The cloud class profiles were downsampled by a factor of four by randomly picking one bin in contiguous sets of four 2B-CLDCLASS height bins, followed by nearest-neighbor interpolation to the target altitude level. The random subsampling was performed to retain the uncertainty introduced by the subsampling of the vertical resolution of the reference data. For the regridding of TIWC, the vertical profiles were first smoothed with a Gaussian filter with full-width at half-maximum of approximately $1\,\mathrm{km}$. Afterwards, the smoothed values were linearly interpolated to the target altitude levels.

Finally, TIWC values were scaled to ensure that the regridded profiles integrate to the same value as the corresponding TIWP.

The vertically subsampled reference data was collocated with the input data by binning the profiles with respect to the input data grid, and then randomly sampling one profile from the multiple profiles collocated with each input pixel. Randomly choosing a profile retains the uncertainty due to the coarser resolution of the input observations, which is required for this uncertainty to be included in the uncertainty estimates provided by the probabilistic regression retrieval. An additional rep-

resentation of reference TIWP was prepared by taking the average per pixel. This alternative representation was included as a sanity check. While it provides the same estimates of the posterior mean, the retrieved uncertainties are with respect to the footprint-averaged reference data, whose distribution changes with the spatial resolution of the observations. However, since it provides essentially the same information as the randomly-sampled TIWP it is not discussed further here. For the temporal collocation, the reference data was assigned to the closest input data with a maximum difference of $15\,\mathrm{min}$ between the profile

observations time and the reference time of the gridded IR data.

CloudSat measurements are available since mid-2006, but are limited to daylight observations from April 2011 due to a battery anomaly (Nayak et al., 2012). Consequently and for simplicity, only CloudSat data before 2011 was considered in order to minimize the risk of introducing a diurnal bias. Data in 2010 were assigned to a held-out test set, and all other three and a half years of data were used for training, with collocations in the first day of each month allocated to a validation set that

was used to monitor training progress.

Training scenes of $384 \times 384$ pixels were generated from the collocated input and reference data. The process involved randomly selecting a pixel with valid reference data as the starting point and then adding a random zonal offset. This process



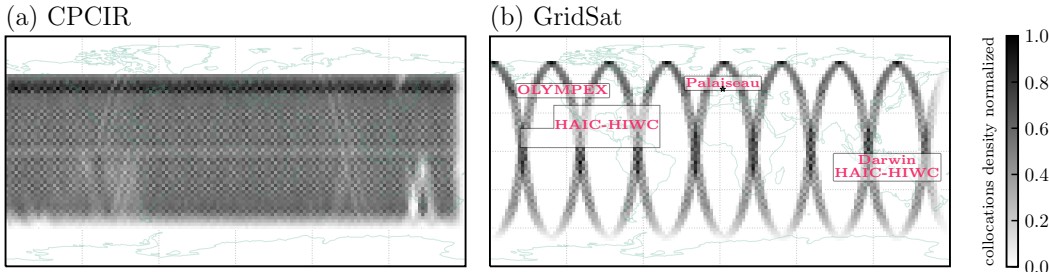

**Figure 1.** Spatial distributions of the pixels with reference values in the training set, binned on a $2.5° × 2.5°$ grid. Panel (b) also displays the approximate extension of three campaigns and the ground-based cloud radar site used in the validation of CCIC.

**Table 3.** Total number of training scenes and pixels with valid profiles for each input dataset in the training dataset.

| Product | Scenes | | | Pixels with reference data | | |
|---|---|---|---|---|---|---|
| | Training | Validation | Test | Training | Validation | Test |
| CPCIR | $3.8 × 10^5$ | $1.5 × 10^4$ | $1.2 × 10^5$ | $1.6 × 10^8$ | $6.5 × 10^6$ | $5.1 × 10^7$ |
| GridSat | $4.1 × 10^4$ | $1.6 × 10^3$ | $1.3 × 10^4$ | $1.7 × 10^7$ | $6.9 × 10^5$ | $5.4 × 10^6$ |
| CPCIR (coarse) | $1.9 × 10^5$ | $7.7 × 10^3$ | N/A | $7.9 × 10^7$ | $3.1 × 10^6$ | N/A |

was repeated until all pixels with valid reference data were included in at least one training scene. Scenes with less than 20% of valid input pixels were discarded. Figure 1 shows that there is a clear difference in the spatial distributions of the collocations

between the two IR data products, which is a result of the fixed overpass times of CloudSat and the lower temporal resolution of the GridSat data. An additional CPCIR training and validation dataset was prepared but at a coarser resolution: the CPCIR data were preprocessed by subsampling every two pixels, thus approximately matching the GridSat resolution, and followed the detailed collocation and scene extraction processes. This aimed to mitigate any potential underfit for GridSat given the imbalanced data; Table 3 shows the counts of the database.

The distributions of the variables (Appendix A) show marginal differences when collocated with GridSat or CPCIR. It is assumed that these differences come from the available collocations for each product (cf. Fig. 1), and are deemed negligible. The distributions of both TIWP and TIWC are heavily right-skewed, spanning several orders of magnitude. Atmospheric states without ice masses are predominant, albeit nearly half of the pixels are cloudy; besides cloud-free scenarios, the three most frequent clouds in the training are altostratus, nimbostratus, and cirrus, in this order (Table A1).

**2.1.3    Neural network architecture**

Figure 2 illustrates the architecture of the CNN used for the CCIC retrieval. The model consists of convolutional encoder and decoder modules that are shared between all retrieval targets and a separate head for every retrieval target. The convolutional blocks used in the en- and decoder of the CNN are similar to those of the ConvNeXt architecture (Liu et al., 2022) with layer





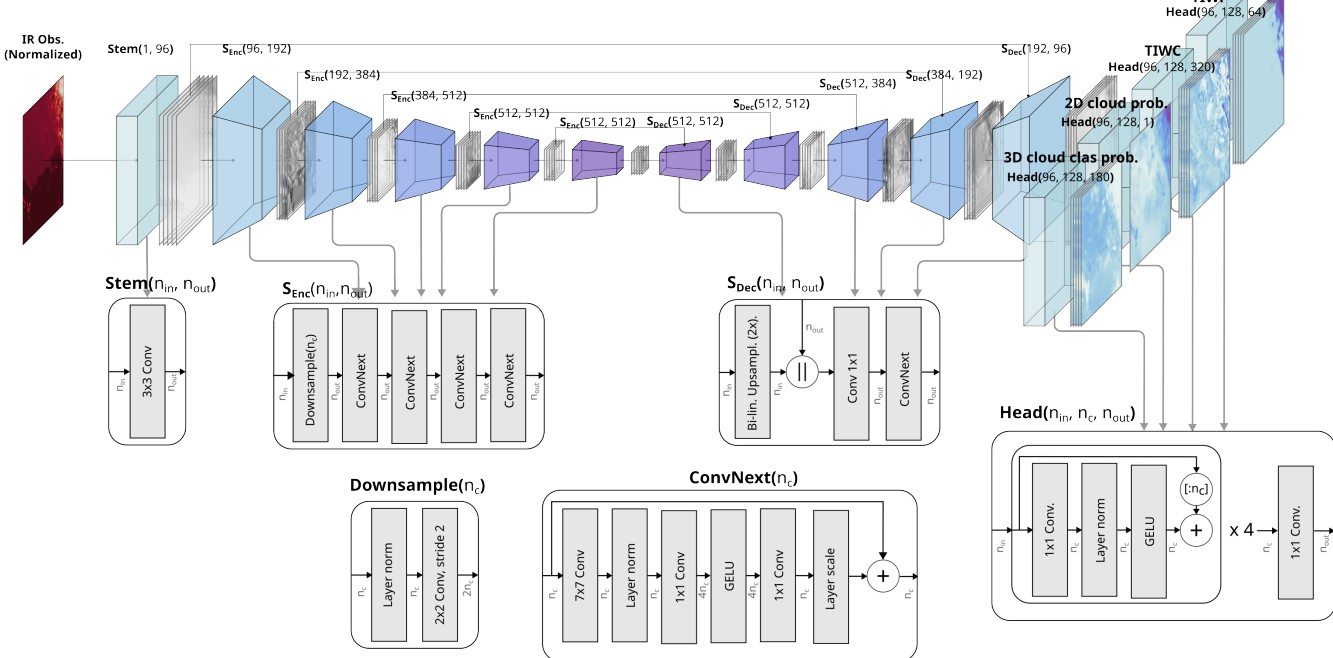

**Figure 2.** The artificial neural network used in CCIC. The input are the brightness temperatures $T_{B,11\,\mu m}$ normalized with $T_{B,11\,\mu m,min} = 170$ K and $T_{B,11\,\mu m,max} = 310$ K. The number of output neurons for each head matches the number of quantiles or classes needed for each variable. The network contains $4.3 \times 10^7$ learnable parameters. Symbols: $||$ – depthwise concatenation; $[: n_c]$ – slicing and keeping the first $n_c$ channels; and $+$ – depthwise addition.

normalization (Ba et al., 2016) arranged in an asymmetric encoder–decoder, U-net-like architecture (Ronneberger et al., 2015).

The network heads each consist of several blocks of $1 \times 1$ convolutions, normalization layers and activation functions and map the shared features extracted by the encoder and decoder to the output of each variable. The basic architecture is grounded in previous retrievals (Amell et al., 2022; Pfreundschuh et al., 2022c, a) and since it provides good results no extensive tuning of the model architecture or hyperparameters was performed.

Because of the very limited information content of the input observations, the mapping from IR brightness temperatures to

hydrometeor concentrations can be expected to exhibit significant retrieval uncertainty. The retrieval uncertainty, which is referred in the machine learning community as *aleatoric uncertainty*, can be quantified using quantile regression neural networks (QRNNs, Pfreundschuh et al., 2018), making the resulting retrieval equivalent to a traditional Bayesian retrieval. Since QRNNs provide a piece-wise linear estimate of the CDF of the marginal distribution of every output variable, no assumptions are required regarding the functional form the distributions. The approach can thus represent non-Gaussian retrieval uncertainties

while simultaneously leveraging the predictive power of deep, convolutional neural networks.



### 2.1.4 Training

The CCIC retrieval was trained using multitask supervised learning, i.e., the losses from all retrieval targets were optimized simultaneously. Since the CCIC retrieval targets comprise both continuous and categorical variables, the network was trained to minimize a mixed loss, defined as the sum of the cross-entropy averages for the categorical variables and the multiple quantile
regression loss averages used in QRNNs for the continuous output variables. No scaling was applied in the total loss function for the different targets. The loss functions for each class of variables are given by

$$\mathcal{L}_{\text{categorical}} = -\frac{1}{N} \sum_{i=1}^{N} \log \left( \frac{\exp y_i^{(c)}}{\sum_{j=1}^{C} \exp y_i^{(j)}} \right) \tag{1}$$

and

$$\mathcal{L}_{\text{continuous}} = \frac{1}{N|\mathcal{T}|} \sum_{i=1}^{N} \sum_{\tau \in \mathcal{T}} (x_i - \hat{x}_i^{(\tau)}) \left( \tau - \mathbb{I}\left( x_i \leq \hat{x}_i^{(\tau)} \right) \right), \tag{2}$$

respectively. Here, $N$ indicates the number of reference values, $y_i^{(j)}$ the output value for class $j$, with class $c$ being the reference class, $\mathcal{T}$ a set of predefined quantiles and $|\mathcal{T}|$ its cardinality, $x_i$ the reference value, $\hat{x}_i^{(\tau)}$ the predicted quantile at level $\tau$, and $\mathbb{I}$ the indicator function.

The continuous variables TIWP and TIWC span over several orders of magnitude, where each order can be considered to contain significant information. The training applied the log-linear transform

$$f(x) = \log(x)\mathbb{I}(x < 1) + (x - 1)\mathbb{I}(x \geq 1) \tag{3}$$

on the reference values, with TIWP and TIWC expressed in $\mathrm{kg\,m^{-2}}$ and $\mathrm{g\,m^{-3}}$, respectively, to address this challenge. The log transform requires positive scalars. This limitation was overcome by replacing values below a clear-sky threshold $t$ using a log-uniform distribution in $[a, b]$: for TIWP (in $\mathrm{kg\,m^{-2}}$), $t = 10^{-3}$, $a = 10^{-6}$, and $b = 10^{-4}$; for TIWC (in $\mathrm{g\,m^{-3}}$), $t = 10^{-7}$, $a = 10^{-10}$, and $b = 10^{-8}$. This treatment of small and zero values helps to have the quantiles of the distribution better
defined as well as also addressing the sensitivity threshold from the measurement instruments. Since the predicted quantiles are invariant to strictly monotonically increasing transformations, the transformation does not change the statistical properties of the resulting prediction.

The QRNN retrievals use different quantile levels for TIWP and TIWC. For TIWP, the network outputs a quantile-parameterized distribution (QPD) with 64 quantile levels, given by $\mathcal{T}_{\text{TIWP}} = \{0.001, 1\Delta_{\text{TIWP}}, 2\Delta_{\text{TIWP}}, \ldots, 62\Delta_{\text{TIWP}}, 0.999\}$, where $\Delta_{\text{TIWP}} =$
$1/63$. For TIWC, the QPD at each altitude level is given by the 16 quantile levels $\mathcal{T}_{\text{TIWC}} = \{0.01, 0.05, 1\Delta_{\text{TIWC}}, 2\Delta_{\text{TIWC}}, \ldots, 13\Delta_{\text{TIWC}}, 0.95, 0.99\}$, with $\Delta_{\text{TIWC}} = 1/15$. This arrangement of the quantile levels was found to produce better retrievals than strictly equally spaced levels. Any quantile between the levels $\mathcal{T}_{\text{TIWP}}$ or $\mathcal{T}_{\text{TIWC}}$ can be obtained with a linear interpolation of the QPD.

The CCIC NN was trained on four NVIDIA Tesla V100-SXM2-32GB GPUs for about 80 epochs using a cosine-annealing
learning rate schedule (Loshchilov and Hutter, 2016), with one epoch taking about $280\,\mathrm{min}$. Monitoring of the validation loss





during training showed no overfitting and the training losses converged. Missing pixels in the input are replaced by a constant special value, while missing reference data is masked out and simply ignored in the loss calculation. Random rotations and flips, followed by random cropping of the input image to $256 \times 256$ pixels, were used to augment the training data. Appendix B provides additional details on the training.

## 2.2 Validation data

The CCIC retrievals are trained to reproduce the data from the CloudSat 2B-CLDCLASS and 2C-ICE datasets, which are themselves derived from remote sensing observations. Since these datasets are used as the ground-truth during training, their errors will be reproduced by CCIC. It is therefore essential to compare the CCIC results to independent and ideally more direct measurements of ice cloud properties. While the most direct measurements of frozen hydrometeors are arguably in-situ measurements, these are inherently sparse and typically only provide estimates of the TIWC at a certain altitude in the atmosphere rather than the TIWP, which would require measuring ice hydrometeor concentrations over the full height of the atmosphere.

In addition to in-situ measurements of frozen hydrometeors, we therefore make use of airborne and ground-based cloud radar observations. Although these measurements will likely be affected by similar uncertainties as the CloudSat-derived reference data, these data allow validating the CCIC retrieval outside the limited temporal sampling of the CloudSat observations. To compare radar observations from the different flight campaigns and the ground-based radar in a consistent manner, we have developed an additional retrieval that retrieves TIWC estimates from radar observations. These retrievals allow us to control the microphysical assumptions used in the retrieval, which constitute a major source of uncertainty in the resulting TIWC estimates. We run the retrieval for multiple assumed ice-particle habits and for each campaign we use the results that yield the best consistency with the in-situ measurements. The implementation of the retrieval is described in detail in Appendix C.

### 2.2.1 HAIC-HIWC

A series of international field campaigns took place between January 2014 and August 2018 to collect in-situ measurements of hydrometeors and other cloud properties: the high altitude ice crystals (HAIC, Dezitter et al., 2013), the high ice water content (HIWC, Strapp et al., 2016a), and the HIWC radar (Ratvasky et al., 2019) projects. These campaigns, referred to as HAIC-HIWC, involved measurements of deep convective systems, including tropical storms, using probes and radar instruments mounted on research aircraft. The hydrometeor total water content (TWC) was one of the properties measured with probes during these campaigns. This property directly correlates with the TIWC provided HIWC conditions, which characterize deep convective weather systems.

The IKP-2 probe (Strapp et al., 2016b; Davison et al., 2016) was specifically developed to measure TWC in HAIC-HIWC. All openly available HAIC-HIWC TWC data measured with this probe were collocated with TIWC CCIC retrievals by averaging them in four-dimensional bins, defined by the GridSat or CPCIR grid, $1\,\mathrm{km}$ altitude bins centered at the CCIC altitude levels, and $30\,\mathrm{min}$ temporal bins centered on the GridSat or CPCIR timestamps. Figure 3 presents the spatial coverage of the collocations obtained with this method.



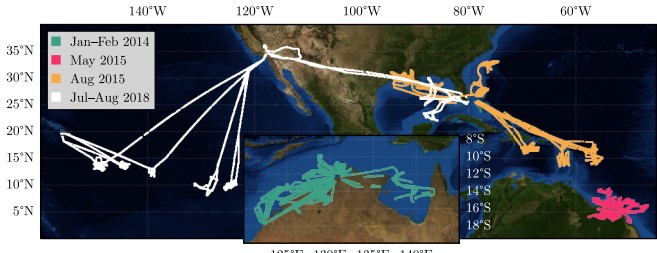

**Figure 3.** Spatial coverage of TWC measured during the different HAIC-HIWC campaigns collocated with CCIC retrievals. The map background is based on NASA Visible Earth imagery.

In addition to the in-situ measurements collected during all flights of the HAIC-HIWC campaigns, the Darwin campaign
also included 95-GHz cloud radar measurements from the Radar Airborne System Tool for Atmosphere (RASTA) radar flown onboard the Falcon 20 of the Service des Avions Francais Instrumentations pour la Recherche en Environnement (SAFIRE). Since these observations allow for a more complete characterization of the vertical structure of the observed clouds than the in-situ measurements alone, they are used here as an additional source of validation data.

### 2.2.2 OLYMPEX

The Olympic Mountains Experiment (OLYMPEX, Houze et al., 2017) was carried out between late fall 2015 and early spring 2016 with the principal aim to investigate the effect of the Olympic mountains on precipitation. As part of the campaign, W-band cloud radar observations were performed by the NASA Cloud Radar System (CRS, Li et al., 2004) on board the NASA ER-2 aircraft. In addition to this, in-situ measurements of concentrations of frozen hydrometeors were performed by the University of North Dakota (UND) Cessna Citation aircraft.

We use both radar-retrievals of TIWC from the NASA CRS radar as well as the in-situ measurements from the UND Citation aircraft to validate the CCIC retrievals. The flight paths of the two aircraft are displayed in Fig. 4. The airborne measurements are collocated with CCIC retrievals by downsampling them to the spatial resolution of the CPCIR and GridSat observations following the method from Sect. 2.2.1.

### 2.2.3 Cloudnet

The Aerosol, Clouds and Trace Gases Research Infrastructure (ACTRIS) Cloudnet data portal curates and provides access to ground-based remote sensing measurements of clouds. In contrast to flight campaigns, measurements from permanent, ground-based radars allow for the evaluation of CCIC over annual and seasonal time scales. For this study we use one year of radar measurements (Delanoë and Haeffelin, 2023) from the Bistatic Radar System for Atmospheric Studies (BASTA, Delanoë et al., 2016) from the site in Palaiseau, France.




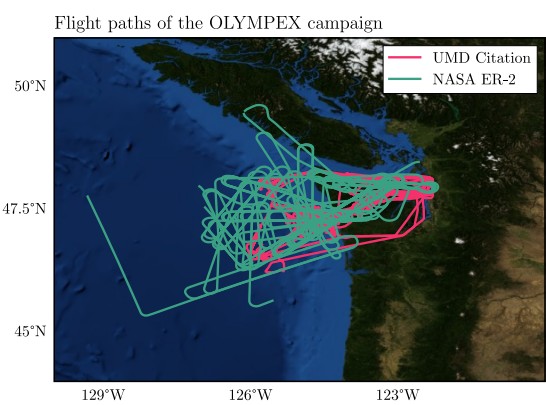

**Figure 4.** Flight paths of the UND Citation and NASA ER-2 aircraft during the OLYMPEX campaign. The map background is based on NASA Visible Earth imagery.

## 3 Retrieval characterization

This section evaluates the accuracy of the CCIC retrieval against held-out test data derived from CloudSat observations.

### 3.1 Case study

To provide an overview of the capabilities and retrieval targets of CCIC, a case study of a mid-latitude cyclone over the West Coast of North America on 3 January 2019 is shown in Fig. 5. The case was chosen as it coincides with an overpass of CloudSat over the cyclone and thus allows comparing the CCIC retrievals to the corresponding CloudSat measurements.

Multiple cloud systems associated with fronts generated by the cyclone can be identified easily even in the IR observations. Compared to the IR input observations, the TIWP field exhibits significantly lower spatial variability, which is expected due to the smoothing effect of retrieval uncertainty on the retrieved posterior mean. Nonetheless, the overall spatial structure of the cloud systems is well reproduced in the TIWP field. It is notable that the distribution of TIWP within the clouds does not seem to exhibit a direct relationship with the corresponding brightness temperatures, indicating that the retrieval leverages spatial context to infer the retrieved TIWP. The contour lines showing the dominant cloud types clearly distinguish stratiform and convective regions of the observed cloud systems whose locations are consistent with the corresponding cloud processes. In particular, stratiform clouds precede the leading warm front of the cyclone, whereas the convective clouds align with the cold and the occluded front. Interestingly, there is also a region of clouds in the South-West corner of the domain, where CCIC identifies St but retrieves no TIWP. This indicates that the retrieval also has capabilities to distinguish the cloud phase.

Compared to the 2C-ICE retrieval along the CloudSat ground track, the CCIC TIWP retrieval exhibits less spatial variability but in general agrees with the 2C-ICE results. The reduced spatial resolution of the CCIC retrieval is even more apparent in the comparison of the retrieved TIWC. Nonetheless, to first order, the vertical structure of the cloud systems is reproduced correctly



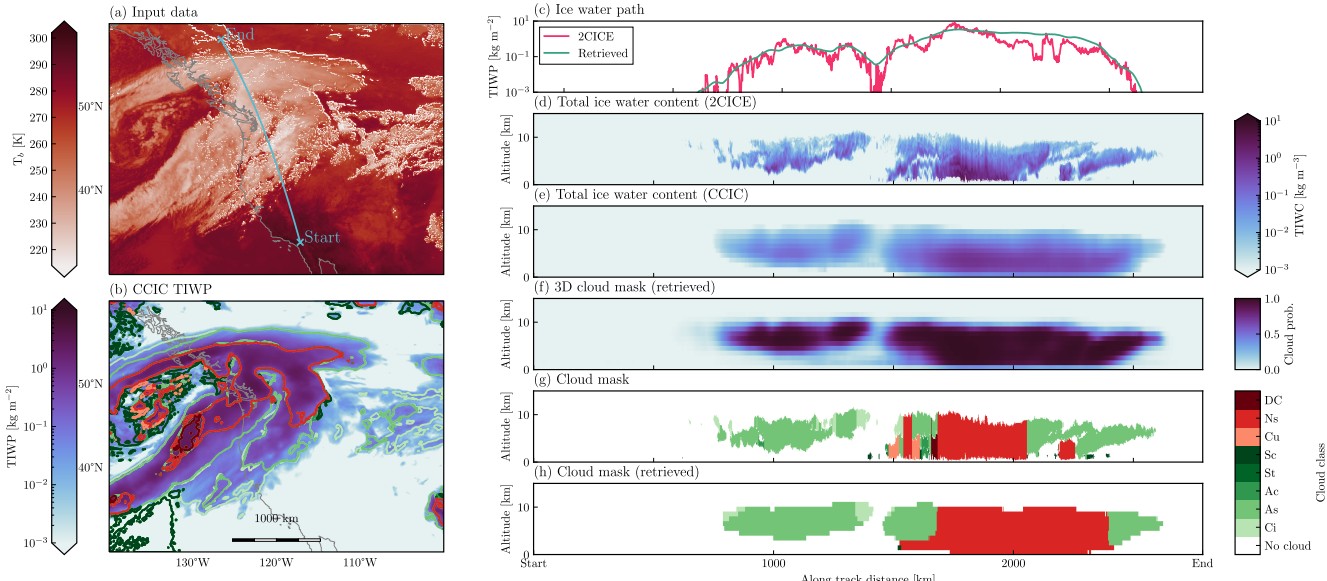

**Figure 5.** Retrieved and reference cloud properties from a CloudSat overpass over a mid-latitude cyclone over the North-American West Coast on 3 January 2019. Panel (a) shows the CPCIR input observations. Panel (b) displays a map of the retrieved TIWP over the region as well as the dominant cloud type, which is defined as the most frequent non-clear cloud class (as defined in panels (g) and (h)) in the atmospheric column. Panel (c) shows the retrieved and reference TIWP along the CloudSat ground track marked by the blue line in Panel (a). Panel (d) shows the TIWC from 2C-ICE along the CloudSat ground track. Panel (e) shows the corresponding retrieved TIWC. Panel (f) shows the retrieved 3D cloud mask. Panel (g) shows the cloud classification from the 2B-CLDCLASS product. Panel (h) shows the corresponding retrieved cloud classes.

by the CCIC TIWC retrieval. In particular, the retrieval correctly reproduces the higher and less-dense clouds associated with the warm front and the lower and denser clouds of the occluded front.

The 3D cloud classification shows general agreement with the classes identified by 2B-CLDCLASS except for the region between 2100 and 2250 km along the transsect. In this region, the full cloud system is classfied as Ns whereas CloudSat detects only a low-level Ns cloud covered by non-convective higher-level clouds. The inability of the CCIC retrievals to resolve the vertically heterogeneous structure of the clouds is certainly due to the limited information content in the IR input observations, which are primarily related to the cloud top temperature. However, an additional factor contributing to the differences between the cloud type classification of CloudSat and CCIC may be that the CCIC cloud-type estimates are from 21:00 UTC, while the CloudSat overpass was close to 21:15 UTC. Since the cloud system can be expected to move to the East, the relative position CloudSat ground track at 21:15 UTC may be shifted slightly to the West, where the cloud cover is more heterogeneous. The cloud classification also fails to reproduce the small-scale variability of the CloudSat-based classification. Nonetheless, it is notable that the retrieval can reproduce much of the vertical structure of the observed cloud system.





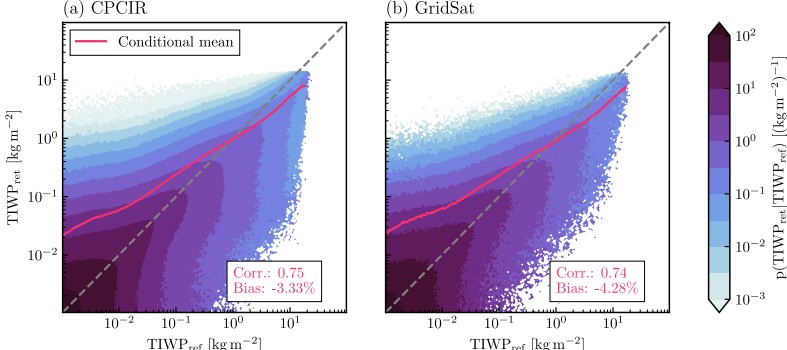

**Figure 6.** Conditional distributions of retrieved TIWP conditioned on reference TIWP. Left panel shows the distribution for the CPCIR input observations; right panel shows the corresponding distributions for the GridSat dataset. The displayed bias and correlation coefficients are computed using all test samples including those outside the range of the scatter plot.

## 3.2 Accuracy on test set

### 3.2.1 Hydrometeor concentrations

Conditional distributions of retrieved posterior mean of the TIWP conditioned on the reference TIWP value are shown in Fig. 6. The logarithmic shading of the distributions reveals large spread about the diagonal indicative of considerable retrieval

uncertainty. Nonetheless, with values of $-3.33\,\%$ and $-4.28\,\%$, the overall retrieval biases are small and the correlations of 0.75 and 0.74 for the CPCIR and GridSat input data, respectively, indicate good sensitivity to TIWP. For either input dataset, the conditional mean of TIWP is biased high up to $1\,\mathrm{kg\,m^{-2}}$ above which it is increasingly biased low.

In addition to the posterior mean estimate of TIWP, CCIC provides a random sample from the posterior distribution together with a $90\,\%$ credible interval (CI). The advantage of random samples from the retrieval posterior distribution is that they provide

a better representation of extreme values than estimates relying on the posterior mean. To illustrate this, Fig. 7 shows the zonal distributions of reference and retrieved TIWP values. The retrieval accurately reproduces the zonal variations of the observed TIWP values both in terms of the mean as well as the distribution of extreme values.

Conditional distributions of the retrieved posterior mean TIWC conditioned on reference TIWC at all altitudes are shown in Fig. 8. The distributions exhibit even larger spread than for the TIWP (Fig. 6), which is expected considering the larger

number of degrees of freedom compared to TIWP. Nonetheless, the retrieval achieves low biases and a correlation coefficient exceeding 0.6, thus demonstrating significant skill in reproducing also the vertical distribution of hydrometeors.

The retrieval biases from TIWP and TIWC show small differences, in the order of a few percent. However, those biases are likely negligible compared to other uncertainties affecting the retrievals. The retrieved TIWC can also be compared directly to the retrieved TIWP by integrating vertically the TIWC. Using a trapezoidal integration and for either input data, the correlation

results 1, the overall bias is at most $2.52\%$, and the conditional mean virtually follows the identity line. Surprisingly, the



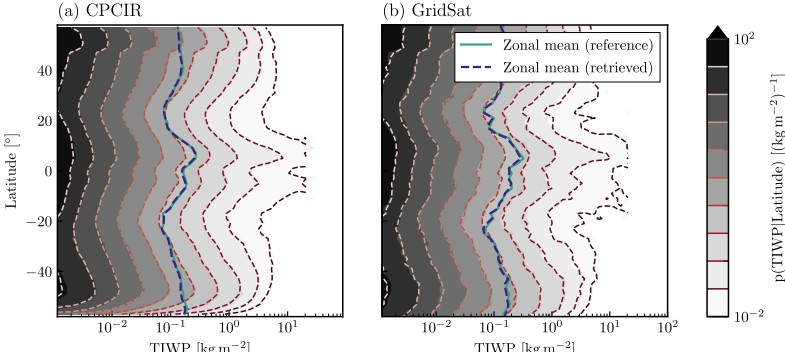

**Figure 7.** Zonal distributions of retrieved TIWP. Filled contours in the background shows the conditional probability density function (PDF) of the reference data. Drawn on top are contours lines of the PDF of random samples of the retrieved posterior distribution of TIWP. The contour levels were chosen so that they correspond to the boundaries between the contour levels of the reference data PDF. Line plots show the zonal mean of reference and retrieved TIWP.

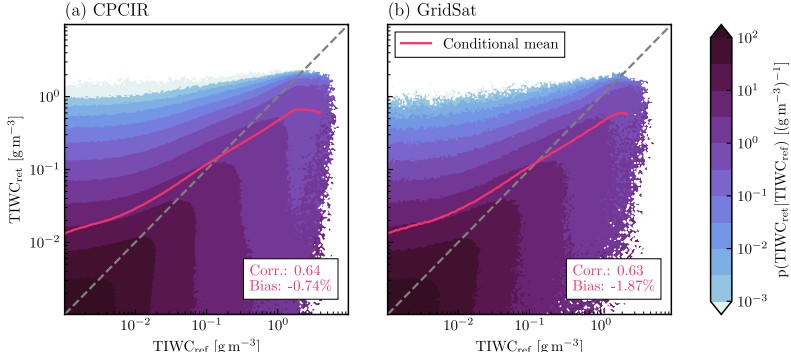

**Figure 8.** Conditional distributions of retrieved TIWC conditioned on reference TIWC. Left panel shows the distribution for the CPCIR input observations; right panel shows the corresponding distributions for the GridSat dataset. The displayed bias and correlation coefficients are computed using all test samples including those outside the range of the scatter plot.





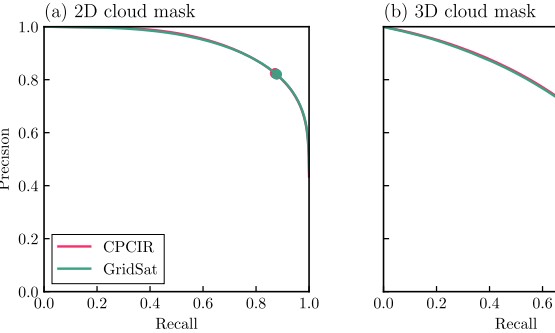

**Figure 9.** Precision-recall curves for the detection of clouds in 2D and 3D. Panel (a) shows the precision and recall for both input-data types for the detection of cloudy columns. Panel (b) shows the corresponding precision and recall for the 3D cloud mask. Circular markers indicate the PR values with the optimal probability threshold in Table 4.

integrated TIWC agrees better with the reference TIWP when comparing the overall biases ($-0.72\%$ and $-1.90\%$ for CPCIR and GridSat) and yields the same correlation values. Nonetheless, the integrated TIWC shows a larger retrieval spread, thereby suggesting a weaker performance as a proxy for retrieving TIWP than the direct retrieval of TIWP.

### 3.2.2 Cloud detection and classification

CCIC provides probabilistic 2D and 3D cloud masks. The 2D cloud mask corresponds to an estimated probability of a cloud being detected by the 2B-CLDCLASS product anywhere in the atmospheric column. The 3D cloud mask classifies all levels in the atmosphere into non-cloudy or any of the eight cloud types distinguished by the CloudSat 2B-CLDCLASS product.

Figure 9 assesses CCIC's skill in detecting clouds in 2D and 3D. The plots display the precision and recall (PR) curves for both input-data types. These curves display the trade-off between the precision, i.e., the fraction of correctly identified clouds 310 and the total number of cloud detections, and the recall, i.e., the fraction of correctly identified clouds and the total number of actual clouds, as the probability threshold above which a cloud is counted as detected is varied. The circular markers show the precision and recall for the optimal probability threshold that was determined as the point on the PR curve that is closest (in terms of Euclidean distance) to the point representing a precision and recall of 1. In order to avoid leakage of information from the test data, the optimal decision thresholds were determined using PR curves calculated on the validation data. The 315 corresponding probability thresholds and precision and recall values are listed in Table 4.

For the detection of clouds in 2D the retrieval achieves a precision and recall in excess of 0.8 indicating good detection skill. For 3D, the detection accuracy decreases yielding a precision of 0.67 and a recall of around 0.74. Again, the decrease in accuracy for the vertically-resolved retrievals is expected due to the higher number of degrees of freedom of the vertically-resolved retrieval targets. Nonetheless, these precision and recall values still indicate notable skill in reproducing the horizontal 320 and vertical distribution of clouds in the atmosphere.





|  | 2D cloud mask | | | 3D cloud mask | | |
|---|---|---|---|---|---|---|
| Input data type | Prob. threshold | Precision | Recall | Prob. threshold | Precision | Recall |
| CPCIR | 0.448 | 0.824 | 0.872 | 0.360 | 0.671 | 0.739 |
| GridSat | 0.440 | 0.820 | 0.878 | 0.364 | 0.670 | 0.737 |

**Table 4.** Optimal probability thresholds determined with the validation set and corresponding precision and recall for the detection of clouds in 2D and 3D for the test set.

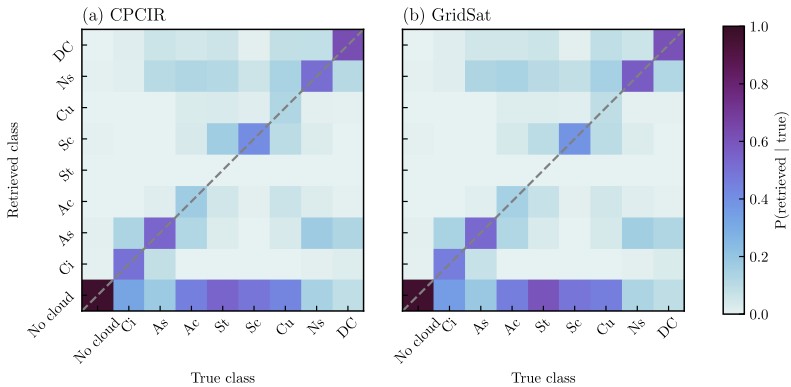

**Figure 10.** Confusion matrix for the 3D cloud classification. Each matrix shows the conditional probability of the retrieved class given a certain true class. Cloud classes use the acronyms from Table 1.

The ability of the retrieval to differentiate between the cloud types of the 2B-CLDCLASS product is assessed in Fig. 10, which shows the confusion matrices for each of the two input datasets. The confusion matrix has been normalized to show the conditional probabilities of the retrieved classes given the reference class. High values in the bottom row indicate that all cloud classes have a relatively high chance to be missed. Nonetheless, for all cloud classes except the Cu and the St class, the
highest probability is located on the diagonal indicating the retrieval is able to identify those clouds. The retrieval is incapable, however, of distinguishing St from Sc clouds and is more likely to assign a St cloud as Sc than correctly detecting it. The same is true for Cu clouds, which the retrieval is likely to mis-classify as Sc, Ns or DC. It is worth noting, however, that both the St and Sc classes are very rare in the reference data distribution and thus highly unlikely a priori. The imbalanced a priori will lead to biases towards the more likely cloud classes in the conditional distributions.

## 4 Validation

The evaluation of the retrieval accuracy on the test data presented in the previous section showed that the CCIC retrieval is able to reproduce the CloudSat reference measurements relatively well. However, the scientifically more relevant question is whether the retrieval can provide reliable estimates even in comparison to independent measurements. If CCIC can reliably





characterize distributions of frozen hydrometeors even outside the temporally and spatially limited sampling of the CloudSat
reference measurements, CCIC would be able complement the CloudSat data record by providing spatially and temporally
continuuous measurements around much of the globe albeit at reduced resolution and accuracy. To investigate this question,
this section compares CCIC retrievals to measurements of cloud hydrometeors from several flight campaigns and ground based
cloud-radar measurements.

## 4.1 HAIC-HIWC

### 4.1.1 Case study

A case study comparing airborne radar and in-situ measurements from the Darwin campaign of the HAIC-HIWC project and
CCIC retrievals is shown in Fig. 11. The flight probed the anvil and core of a deep convective system in the Gulf of Carpentaria.
In the beginning of the flight, the aircraft passed through low-level clouds while still close to Darwin. The aircraft entered the
anvil of the system at around 21:00 UTC and then passed through the core of the system several times. Overall, both CCIC
retrievals capture the horizontal and vertical extent of the observed cloud fairly well. However, the CPCIR-based retrieval
overestimates the horizontal extent of the anvil cloud. A potential explanation for this is that during the time of the campaign,
the CPCIR input data around Darwin is missing at the full hour and the results at 21:00 are thus linearly interpolated between
the results at 20:30 and 21:30. In contrast to that, the GridSat-based retrieval, which has valid input observations at 21:00 UTC,
reproduces the horizontal extent of the anvil cloud more accurately.

The CCIC retrievals are not able to capture the internal structure of the clouds and underestimate the variability in the
magnitude of the TIWC between the anvil and the core of the convective system. To a certain extent this is expected considering
the very limited information provided by the single channel IR observations used as input from the retrieval. Nonetheless, it is
notable that the CCIC retrieval, at least to first order, yields estimates of TIWC that are consistent with both the radar and the
in-situ measurements.

## 4.2 HAIC-HIWC: aggregated data

For a broader comparison of the CCIC retrievals and the campaign measurements, Fig. 12 shows the distributions of retrieved
TIWC conditioned on in-situ measurements from all HAIC-HIWC campaigns. The CCIC retrievals exhibit a tendency to
overestimate TIWC at concentrations below $0.1\,\mathrm{g\,m^{-3}}$ and underestimate it above $1\,\mathrm{g\,m^{-3}}$. Overall, however, the CCIC-
retrieved TIWC is reasonably well correlated with the HAIC-HIWC in-situ measurements across all campaigns of the project.

Scatter plots of CCIC-retrieved TIWC and TIWP conditioned on reference retrievals from the RASTA airborne cloud radar
used during the Darwin campaign are shown in Fig. 13. The radar retrievals for the HAIC-HIWC campaign use the Large
Column Aggregate particle, which yielded the best agreement with collocated in-situ measurements (cf. Appendix C). With
respect to TIWC, the overestimation and underestimation tendencies are consistent with the validation against the in-situ
measurements, and the retrieved values also are reasonably well correlated against the reference values. In terms of TIWP,
the CCIC retrievals are biased high for reference values below $1\,\mathrm{kg\,m^{-2}}$ but the overall bias remains comparably low. With



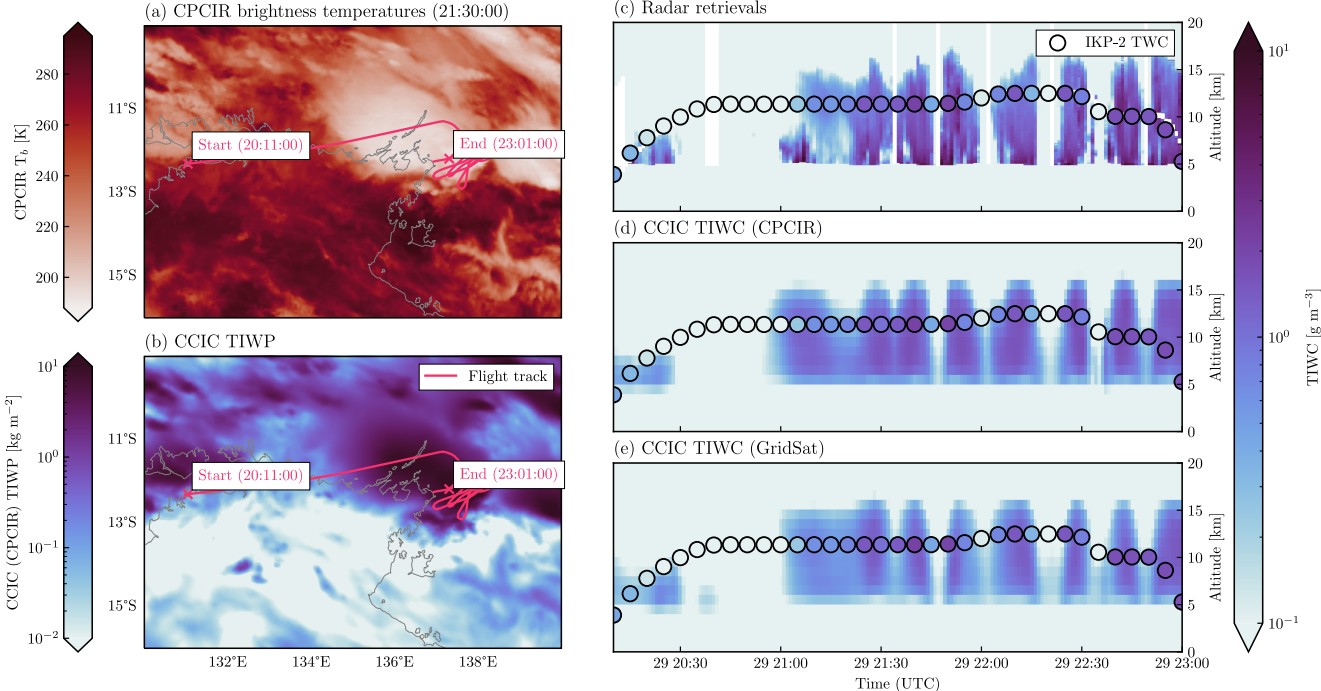

**Figure 11.** In-situ measurements and retrieval results for flight 10 of the Darwin HAIC-HIWC campaign on 29 January 2014. Panel (a) shows the CPCIR 11 µm brightness temperatures. Panel (b) shows the corresponding CCIC TIWP field. Panel (c) shows the TIWC retrieved from the RASTA cloud radar using the Large-Column Aggregate as ice particle shape. Circular markers show the five-minute average in-situ-measured TWC from the IKP-2 probe. Panels (d) and (e) show the TIWC retrieved from CPCIR and Gridsat observations and interpolated to the flight path.

values of 0.66 and 0.62 for the CPCIR and GridSat-based retrievals, respectively, the correlations are higher than for the TIWC estimates.

## 4.3 OLYMPEX

A case study depicting CCIC retrieval results together with in-situ measurements and TIWC derived from radar observations
from the NASA CRS is shown in Fig. 14. The depicted flight is the one that allowed for the largest number of collocations between radar retrievals from the NASA CRS system on the ER-2 aircraft and the in-situ probes on the UND citation aircraft. During the flight the two aircraft profiled a complex prefrontal storm over the Olympic Peninsula. The radar retrievals show the vertical extent of cloud extending up to 8 km and decreasing down to 3 km towards the end of the flight leg. The CCIC retrieval from the CPCIR observations capture the overall shape of the cloud system but overestimate its vertical extent. The
GridSat-based retrieval is even less successful in reproducing the extent of the cloud. This is likely due to the lower temporal resolution of the input data, which required interpolation between the available results at 15:00 and 18:00 UTC.




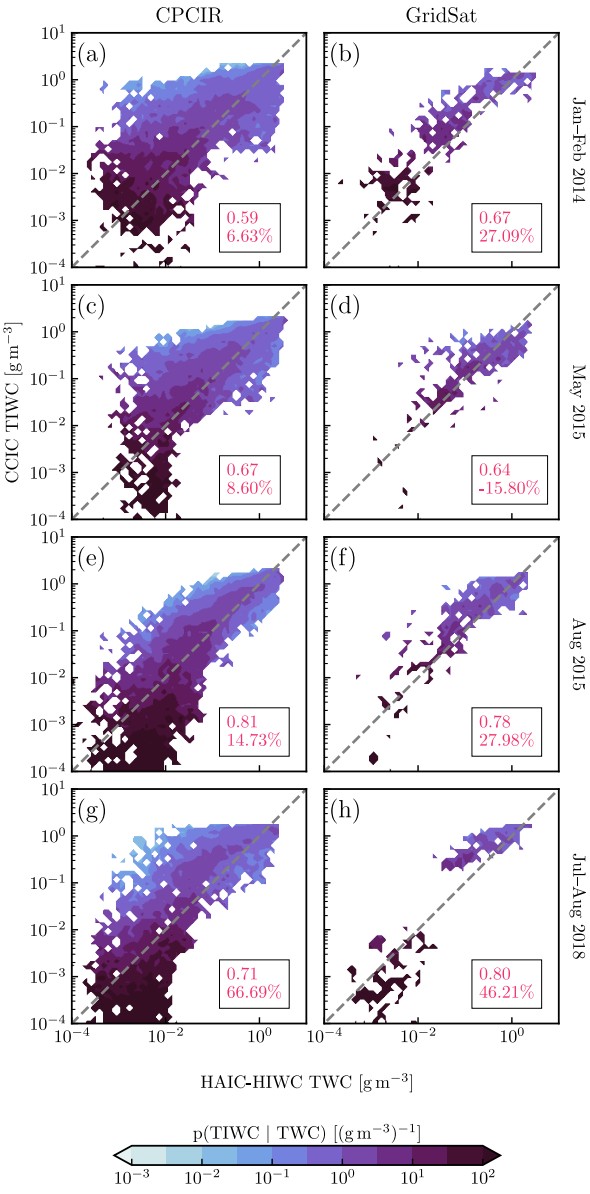

**Figure 12.** Distributions of retrieved TIWC conditioned on reference TIWC from the in-situ measurements for each of the HAIC-HIWC campaigns presented in Fig. 3. The left column contains CPCIR retrievals, the right column GridSat retrievals; the first number in the box indicates the correlation, the second the bias. The displayed bias and correlation coefficients are computed using all test samples including those outside the range of the scatter plot.

The radar observed broken clouds from about 16:25 to 16:28 while the in-situ measurements exhibit spatially more continuous hydrometeor concentrations. The broken clouds observed by the radar indicate high spatial and temporal variability in





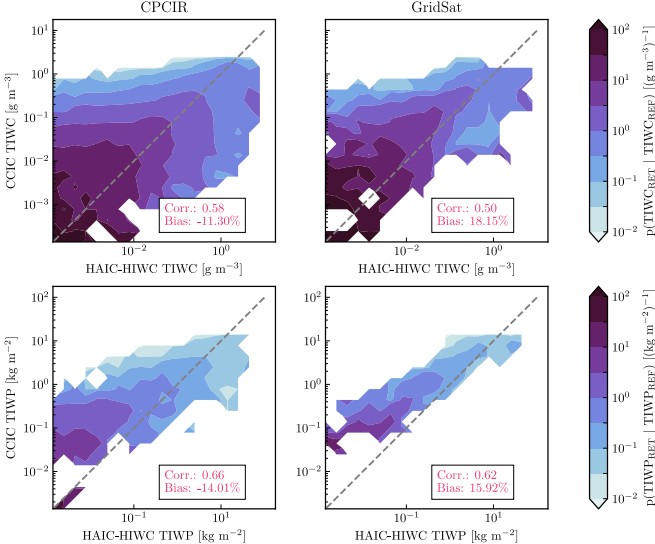

**Figure 13.** Distributions of retrieved TIWC (top) and TIWP (bottom) conditioned on reference values from airborne radar retrievals during the Darwin HAIC-HIWC campaign (Jan–Feb 2014) obtained with the Large-Column Aggregate particle model. The displayed bias and correlation coefficients are computed using all test samples including those outside the range of the scatter plots.

the cloud field that is too high for the CCIC retrievals to resolve thus leading to the overestimation of the vertical extent of the
cloud system. It should also be noted that the in-situ measurements indicate the presence hydrometeors even where no cloud is observed in the radar measurements, providing further evidence that deviation between radar, in-situ and CCIC measurements is, at least in part, due to the variability in the cloud field.

Scatter plots of the conditional distribution of retrieved TIWC conditioned on reference TWC and TIWC are displayed in Fig. 15. Compared to the in-situ measurements, the CPCIR CCIC results are biased high and exhibit relatively low correlation
of 0.37. With a value 0.65, the correlations are significantly higher for the GridSat retrievals. Since all other comparisons showed a high degree of similarity between the CPCIR and GridSat-based results, the higher correlation is probably due to the spatial and temporal sampling of the collocations used for the validation of the GridSat-based results. The GridSat-based retrievals also exhibit a weaker bias in the TIWC estimates. The correlations with respect to the airborne radar measurements are significantly higher with values of 0.6 for the CPCIR-based retrievals and 0.73 for the GridSat-based retrievals. The biases
with respect to the radar measurements are similar to those observed for the in-situ measurements. The radar results shown here were obtained with the Large-Plate Aggregate particle model. Although the few available collocations between radar and in-situ measurements showed the best agreement for the 8-Column Aggregate particle model, the Large-Plate Aggregate led to better agreement between the biases between radar estimate and CCIC and in-situ measurements and CCIC.

The conditional distributions show that the CCIC retrievals severely overestimate TIWC for low reference concentrations
in some cases. The temperature contours indicate that these cases of overestimation are associated with comparably high temperatures and thus correspond to estimates lower down in the atmosphere, where the input observations provide little





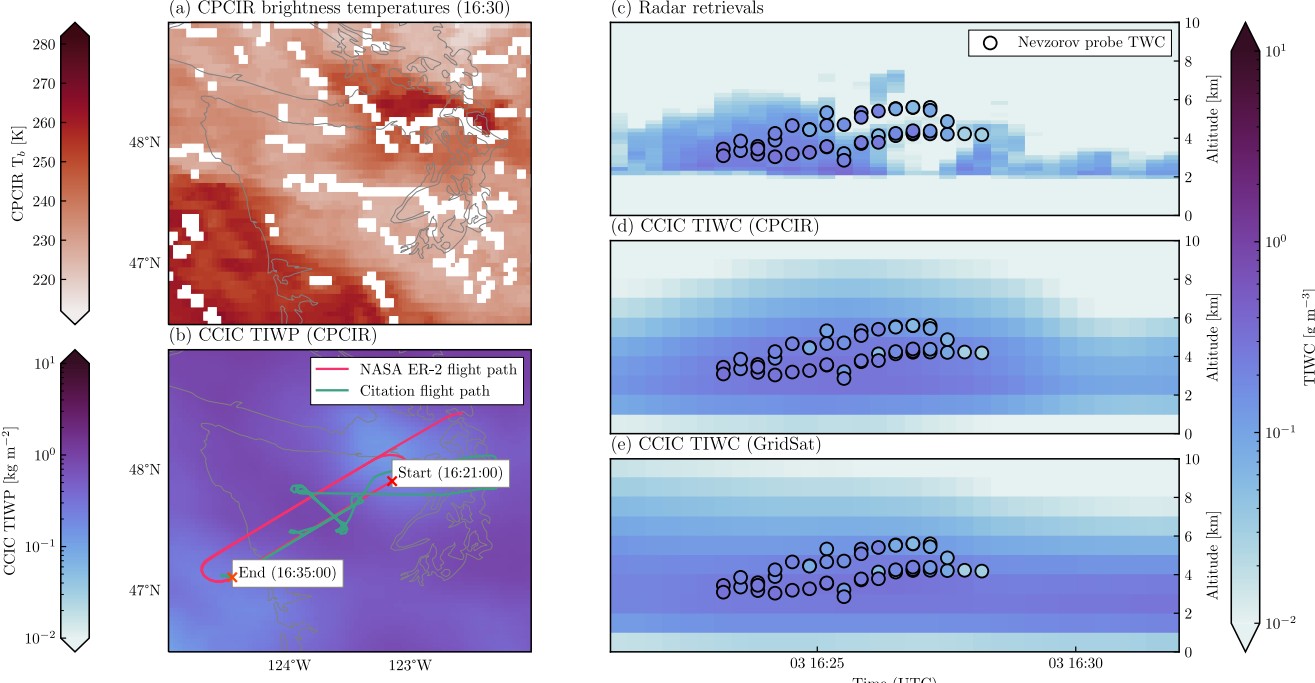

**Figure 14.** Collocated in-situ, radar, and CCIC measurements of ice hydrometeors from a flight on 4 December 2015 during the OLYMPEX campaign. Panel (a) shows the CPCIR input data for the CCIC retrieval over the Olympic Peninsula. White pixels mark missing values in the input data. Panel (b) shows the corresponding retrieved TIWP. Panel (c) shows the TIWC retrieved from the NASA CRS on the ER-2 aircraft using the Large-Plate Aggregate ice particle model. Scatter points show collocated Nevzorov-probe measurements of TWC from the UND Citation aircraft within $5\,\mathrm{km}$ and $15\,\mathrm{min}$ of the radar observations. Panel (d) shows TIWC along the flight path derived from CPCIR input data. Panel (e) shows the TIWC derived from GridSat observations.

information to guide the retrieval. Moreover, the limited vertical resolution and lack of input data constraining the thermal structure of the atmosphere may lead to TIWC being retrieved close to and even below the freezing level.

Finally, scatter plots showing the distributions of retrieved TIWP conditioned on the radar-retrieved TIWP are shown in
Fig. 16. Although the spread in the distributions is reduced, the correlation with the reference measurements decreases for the CPCIR retrievals. The reduction in correlation for the TIWP retrievals is somewhat surprising considering that TIWP should be easier to retrieve due to the lower number of associated degrees of freedom. What may contribute to the reduced correlation, however, is the relatively low sensitivity of the radar retrievals, which is evident in the lack of reference measurements of TIWP values below $10^{-2}\,\mathrm{kg\,m^{2}}$.



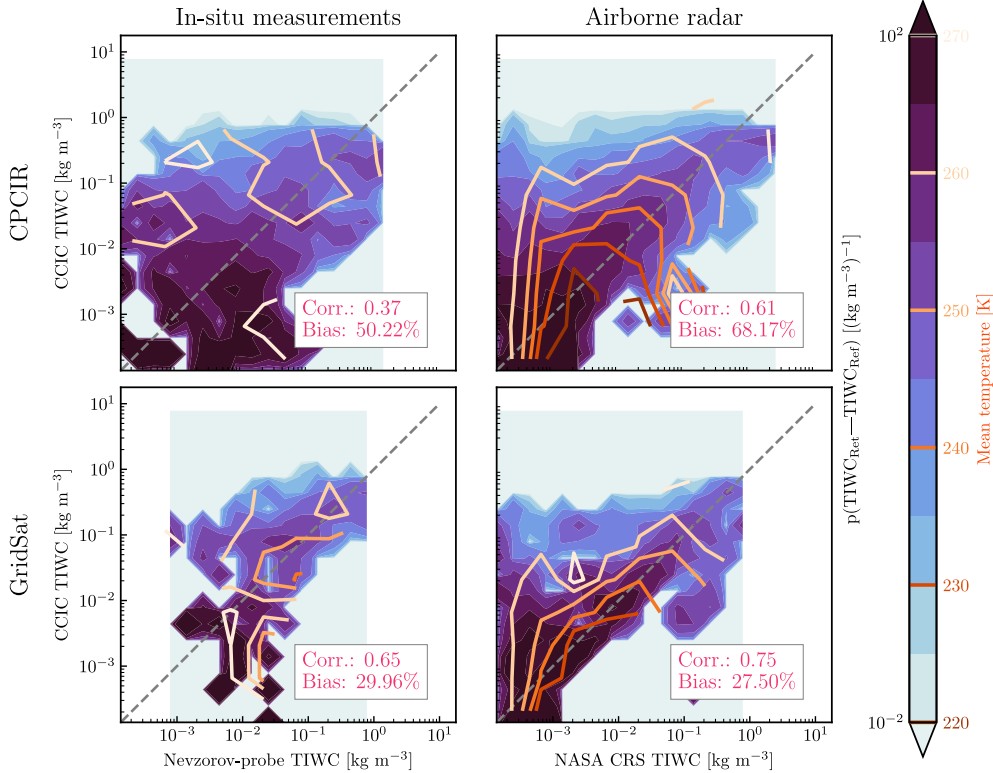

**Figure 15.** Distributions of retrieved TIWC conditioned on reference TIWC measurements during the OLYMPEX campaign. Contour lines show the mean of the ERA5-derived ambient temperature for the retrieval results in each bin. The displayed bias and correlation coefficients are computed using all test samples including those outside the range of the scatter plots.

## 4.4 Cloudnet

Fig. 17 compares CCIC retrievals with TIWC and TIWP estimates derived from the Cloudnet ground-based cloud radar in Palaiseau. The conditional distributions of collocated TIWP and TIWC exhibit similar spread as for the flight campaigns. The correlations are slightly lower than those obtained for the flight campaign data. The CCIC retrievals slightly overestimate hydrometeor concentrations compared to radar retrievals using the Large Plate Aggregate particle model.

Figure 18 shows diurnal cycles of TIWP retrieved from the ground-based radar and the CCIC retrievals. Calculated over the full year 2019, the CCIC retrievals correctly reproduce the early-morning peak observed in the radar measurements although they underestimate its strength. The CCIC retrievals still capture most changes in the diurnal variations on seasonal time scales. The exception here are the summer months, during which the CCIC results only show a very weak early-morning peak, whereas it is fairly pronounced in the radar data. This seems to indicate that the retrieval uncertainty in individual events is too large to capture diurnal variations reliably over the course of a single season. Due to their lower temporal resolution, the GridSat-based results generally exhibit weaker variations than the CPCIR retrievals.





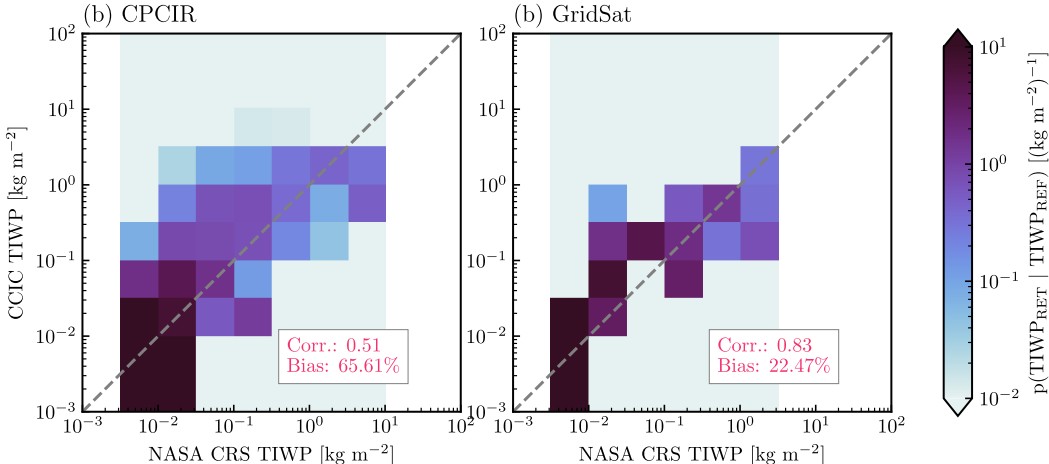

**Figure 16.** Distributions of retrieved TIWP conditioned on reference TIWP measurements during the OLYMPEX campaign. The displayed bias and correlation coefficients are computed using all test samples including those outside the range of the scatter plots.

Overall, it is notable that the CCIC retrievals manage to reproduce the diurnal and seasonal variation relatively well. It is important to note that CloudSat measurements are essentially limited to two discrete local overpass times due to the sun-synchronous orbit of the satellite. Therefore, they cannot resolve the diurnal cycle of cloud properties. The good agreement with the ground-based measurements shows that, despite being based on CloudSat measurements, the CCIC retrieval can reproduce diurnal variations in TIWP. The CCIC retrievals thus have the potential to provide an important novel perspective on ice clouds in the atmosphere.

## 5 Discussion

This study introduced the CCIC ice cloud retrieval that produces estimates of TIWP, TIWC, and cloud coverage from single-channel geostationary IR observations. The input data was deliberately restricted to a single channel of geostationary observations so that the retrieval can be applied to the entire historical record of geostationary satellite observations. To demonstrate the soundness of the concept and its implementation, we have presented a thorough assessment and validation of the neural-network based retrieval.

### 5.1 Retrieval accuracy

The assessment of the retrieval accuracy on the held-out test dataset (Sect. 3) showed that the CCIC retrieval can reliably reproduce the CloudSat 2C-ICE estimates although individual retrievals may exhibit considerable uncertainty. CCIC can even reproduce the vertical distribution of ice hydrometeors despite its input being limited to a single channel of IR observations.



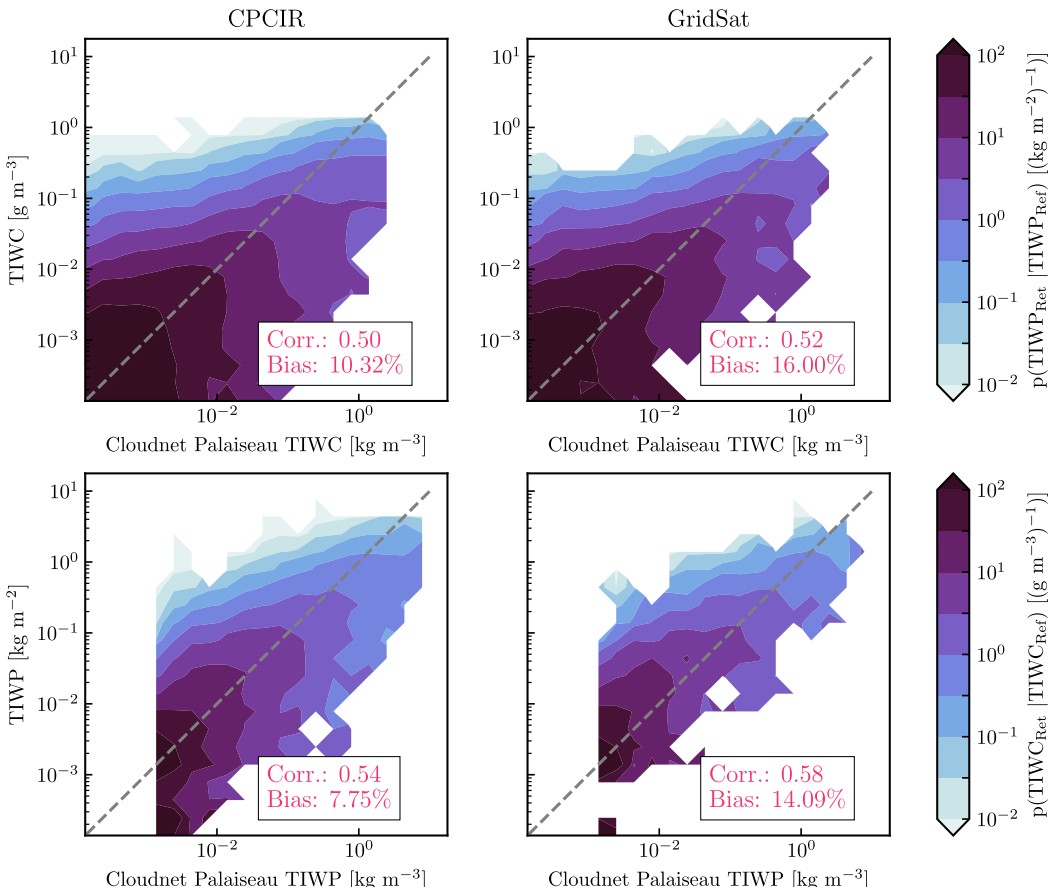

**Figure 17.** Conditional distributions of retrieved TIWP (first column) and TIWC (second columns) retrievals conditioned on reference measurements from the ground-based cloud radar in Palaiseau. The first row shows the results for the CPCIR-based CCIC retrievals. The second row shows the results for the GridSat-based CCIC retrieval. The displayed bias and correlation coefficients are computed using all test samples including those outside the range of the scatter plots.





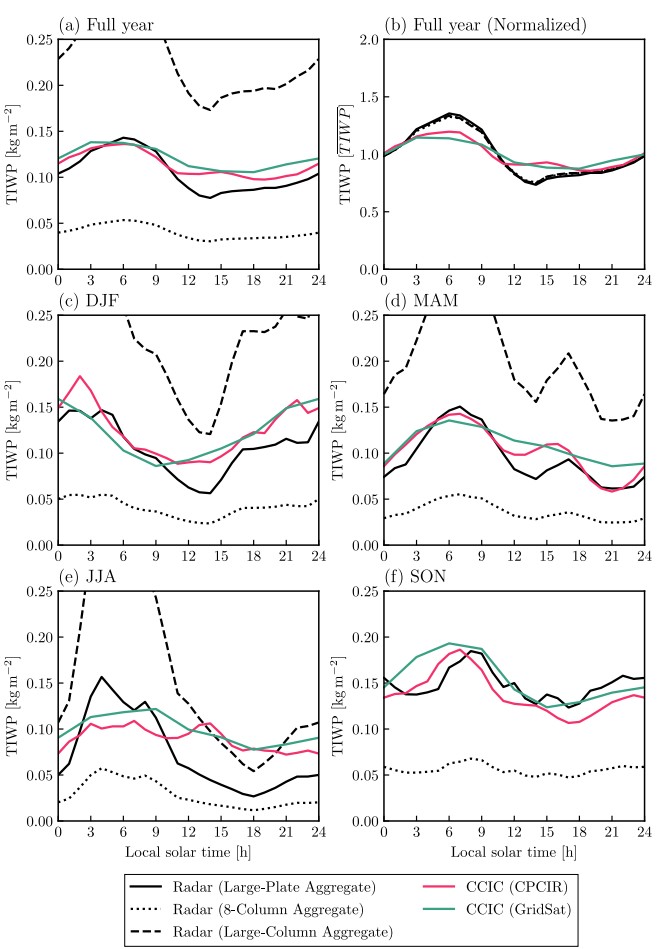

**Figure 18.** Diurnal cycles of TIWP retrieved from CCIC and the Cloudnet ground-based cloud radar in Palaiseau. Panel (a) shows the dirunal cycles in absolute TIWP values. Black lines show the retrieved diurnal cycles for the three different particle models used in the radar retrievals. Panel (b) shows diurnal cycles normalized by the mean TIWP. Panels (c)-(f) show seasonal diurnal cycles.



This is certainly notable, especially since it is commonly understood in the remote sensing community that broader spectral coverage is required to reproduce the vertical structure of clouds.

This study further validated the retrieval against measurements from two flight campaign series and a ground-based cloud radar. The CCIC retrievals generally agree with the radar measurements and most in-situ measurements. The biases with respect to in-situ measurements were within or close to $50\%$ for both flight campaign series, which is an encouraging result considering the overall uncertainty associated with estimates of ice hydrometeors. The lowest sensitivity was found for the CPCIR-based retrievals for the OLYMPEX campaign where the correlation with the in-situ measurements was only 0.37. The

likely reason for this is that the in-situ measurements from the UND citation aircraft mostly sampled altitudes close to the cloud base where the retrieval uncertainties are likely higher. This is confirmed by the much better correlation coefficient found for the comparison to the radar-derived TIWC measurements. The otherwise good agreement between in-situ measurements and CCIC TIWC retrievals confirms the ability of the retrieval to resolve the vertical distribution of ice hydrometeors in the atmosphere.

The validation data covered the time range 2014 – 2020 and climatic regimes from the tropics to the mid-latitude. The fact that the CCIC retrieval yields reliable results even far outside of the period used for training the retrieval (2006 – 2009) constitutes preliminary evidence that the retrieval results are stable in time. Moreover, CCIC exhibits comparable sensitivity (measured in terms of correlation with the campaign measurements) and biases for both tropical and mid-latitude cloud regimes, indicating robustness of the retrieval across climate zones. In general, the CCIC retrievals tend to overestimate in-situ and radar

measurements. This is consistent with a similar tendency to overestimate in-situ-measured hydrometeor concentrations found for the 2C-ICE product (Deng et al., 2010, 2013), which is used as reference data to train the CCIC retrieval.

Overall, the validation results are very encouraging and indicate that CCIC provides reliable results that constrain not only the integrated density of frozen hydrometeors but also their vertical distribution. Moreover, good agreement with the diurnal cycles derived from the ground-based radar shows that CCIC complements currently available CloudSat measurements by

providing retrievals outside their limited spatial and temporal coverage.

Despite these encouraging results, CCIC should still be considered a proof of concept. CCIC's principal objective remains to explore the potential of modern deep-learning techniques to expand the observational climate record of ice clouds. We aim to produce a full climate record of cloud retrievals but acknowledge that the long-term stability of these deep-learning-based retrievals remains an open question, which we aim to address in a follow-up study. Because of the exploratory character of

CCIC (and the limited funding available for the project), there is undoubtedly room to improve the retrieval further. Although based on the state-of-the-art ConvNeXt architecture, the employed neural network architecture has yet to be optimized exhaustively. Further potential opportunities for improving the retrieval are unsupervised pre-training, and incorporating the temporal evolution into the retrieval.

## 5.2 Limitations

Since CCIC uses the 2C-ICE and 2B-CLDCLASS products as reference data, it will directly inherit their characteristics. In particular this means that CCIC is based on the same microphysical assumptions as these two products and will therefore repro-





duce their errors. However, our validation showed that the resulting estimates agree reasonably well with in-situ measurements, thus instilling confidence in the reliability of both the reference data and the CCIC retrievals.

A principal difficulty of measuring concentrations of hydrometeors in the atmosphere is establishing a ground truth. In-situ
measurements are difficult to conduct and therefore rare. Furthermore, available measurements have to be collocated with satellite-based retrievals, whose measurements typically extend over significantly larger measurement volumes. Since both flight campaigns considered here comprised a relatively large amount of flight hours, we were able to average the campaign measurements to the native resolution of the satellite retrievals. While this yielded fairly good agreement, we note that, despite the averaging, the measurement volumes are likely to remain vastly different, which will contribute to the error between
retrieval and validation data.

Quantitative estimates of hydrometeor concentrations derived from radar measurements exhibit significant uncertainty due to their sensitivity to the assumed microphysical properties, which are difficult to constrain a priori. The approach taken here was to perform multiple radar retrievals with different assumptions on the particle shape and constrain the results through comparison with the in-situ measurements. This worked well for the HAIC-HIWC campaign, for which the best agreement
between radar and in-situ measurements was found for the Large Column Aggregate particle model, but lead to inconsistencies in the biases observed between CCIC and the OLYMPEX campaign in-situ measurements and radar retrievals. For OLYM-PEX we therefore chose the Large Plate Aggregate as the most suitable particle model and used the same for the Palaiseau results, where it lead to good agreement with the CCIC retrievals. While the results are reasonably consistent, we note that microphysical assumptions constitute a major uncertainty for the radar retrievals used in this study.

### 5.3 Applications

Given that CCIC uses only single-channel IR observations as retrieval input, an important question to answer is whether CCIC's retrievals add new information that cannot be readily obtained from the input data. To address this question, the following sections discuss two possible applications where the CCIC retrievals may provide an advantage over the original IR brightness temperatures.

### 5.3.1 Cloud tracking

A common application of IR observations from geostationary satellites is the tracking of convective cloud systems. Tracking algorithms typically use IR brightness temperatures to identify cold clouds (Feng et al., 2021; Esmaili et al., 2016; Fiolleau and Roca, 2013). The issue with this is that IR brightness temperatures reflect the location of the cloud top with respect to the thermal structure of the atmosphere rather than the processes forming the cloud. Since the CCIC retrievals can reproduce the
vertical structure of clouds and identify different cloud types, the CCIC results may provide a better way to identify and track cloud systems. While the most natural way of tracking cloud systems would be using the cloud detection outputs provided by CCIC, this approach has the disadvantage of not having a direct correspondence in the output fields from climate models, which is often desired to allow comparison of model and observation-derived tracking databases. However, it seems reasonable to assume that also the TIWP field provides a more direct signal to track clouds than the IR brightness temperatures.





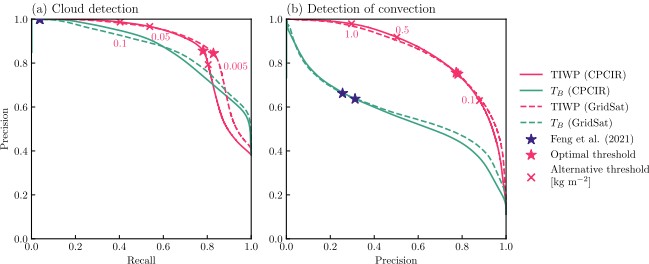

**Figure 19.** Precision-recall curves for the detection of clouds and convection using IR-brightness temperatures and CCIC TIWP estimates for the test set. Panel (a) assesses the ability of the two quantities to identify cloudy 2B-CLDCLASS columns. The blue stars mark resulting performance of the $241\,\mathrm{K}$ threshold used by Feng et al. (2021) to identify cold cloud shields for the CPCIR and GridSat datasets. The pink stars mark the optimal decision threshold determined as the points closest (in terms of Euclidean distance) to a precision and recall of 1. The corresponding threshold values are $0.009\,\mathrm{kg\,m^{-2}}$ for the CPCIR dataset and $0.012\,\mathrm{kg\,m^{-2}}$ for GridSat. Additional markers show precision and recall for selected, additional TIWP detection thresholds. Panel (b) shows the same results for the detection of convective clouds (Cu, Nb or deep convection) anywhere in the column. The blue stars mark the classification accuracy of the $225\,\mathrm{K}$ threshold used by Feng et al. (2021) to identify convective cores. The optimal threshold values for CPCIR and GridSat datasets are $0.18\,\mathrm{kg\,m^{-2}}$ and $0.16\,\mathrm{kg\,m^{-2}}$, respectively.

500 The global meso-scale convective system database by Feng et al. (2021), for example, uses a threshold of $225\,\mathrm{K}$ to identify a deep convective core and tracks the surrounding cold cloud shield consisting of surrounding pixels with brightness temperatures below $241\,\mathrm{K}$. The ability of the raw IR brightness temperatures and the CCIC-retrieved TIWP fields to identify convective clouds and associated cloud shields can be assessed with the test data used in Sect. 3. To this end, we classify an atmospheric column associated with a pixel of CPCIR observations as cloudy and convective based on the corresponding cloud-scenario

505 from the 2B-CLDCLASS product. A pixel is classified as cloudy if the corresponding column contains a cloud at least at one level. Similarly, it is classified as convective if it contains a Cu, Nb, or deep convective cloud anywhere in the column. Fig. 19 shows precision-recall curves for the detection of clouds and the detection of convective pixels using different brightness temperature and TIWP thresholds.

 The TIWP offers slightly better detection skill for detecting clouds, however this is mostly limited to the region where

510 the precision is around 0.8. For comparison, the global database by Feng et al. (2021) uses a threshold of $241\,\mathrm{K}$ to identify cold cloud shields. Since this value is relatively low and unlikely to be produced by anything else than a cloud, the threshold achieves a precision very close to 1 but only a very low recall indicating that it misses a large number of clouds. Larger differences between the brightness temperatures and TIWP are found for the identification of convective cores. Here the TIWP offers a significant advantage compared to the brightness temperature threshold of $225\,\mathrm{K}$ used by Feng et al. (2021). A detection

515 threshold of $\mathrm{TIWP} > 0.18\,\mathrm{kg\,m^{-2}}$ yields slightly higher precision and a recall that is more than two times as high as when brightness temperatures are used.

 A comparison of the identified clouds for the case study depicting the mid-latitude cyclone shown in Fig. 5 is presented in Fig. 20. As these results show, the brightness temperature thresholds fail to identify any of the convective clouds identified



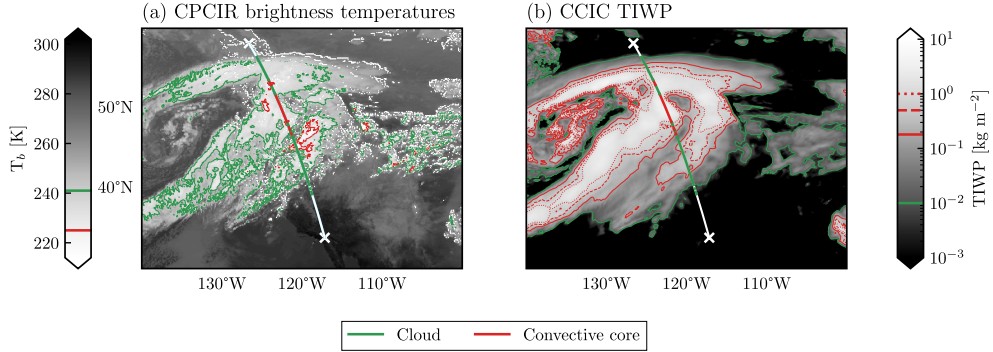

**Figure 20.** Cloud and convective regions as identified in brightness temperatures and TIWP for the mid-latitude cyclone shown in Fig. 5. Panel (a) shows the identified regions based on the brightness-temperature-thresholds used in Feng et al. (2021) together with classes identified along the CloudSat overpass. Panel (b) shows the corresponding regions identified based on TIWP. Solid lines show regions resulting from the optimal detection thresholds identified in 19. Dashed and dotted lines show convective regions identified for higher detection thresholds of $\text{TIWP} > 0.5\,\text{kg}\,\text{m}^{-2}$ and $\text{TIWP} > 1.0\,\text{kg}\,\text{m}^{-2}$, respectively.

by CloudSat. Moreover, the identified cloud field is discontinuous and fails to reproduce the structure of the cyclone. The
TIWP-based classification identifies clouds more reliably and better reproduces the structure of the cyclone but overestimates the extent of the convective clouds associated with the cold front. Overall, the TIWP-based classification provides a better representation of the cloud systems associated with the cyclone that is in fairly good agreement with the general understanding of cloud processes occurring in frontal zones. Moreover, we note that the overestimation of convective regions may be balanced by choosing a higher detection threshold that would achieve a lower recall but higher precision, as exemplified by the dashed
and dotted contours in Fig. 20.

Since the issues of brightness-temperature-based cloud tracking are well known, Feng et al. (2021) and most other cloud-tracking approaches incorporate additional data such as precipitation estimates. Nonetheless, the results shown in Fig. 20 indicate that the features identified in brightness temperature fields may not provide a good representation of actual cloud systems. The retrieved TIWP fields directly estimate the concentration of ice particles in clouds and therefore provide a more
direct signal to identify clouds than the IR brightness temperatures. Furthermore, high TIWP values are likely more directly related to convection than high cloud tops alone. This reasoning together with the results shown in Fig. 20 suggests that cloud tracking on TIWP likely provides a better representation of actual cloud systems than brightness-temperature-based trackings.

### 5.3.2 Climate studies and model assessment

Compared to raw brightness temperatures, TIWP has the advantage of being directly comparable to physical quantities repre-
sented in weather and high-resolution climate models. This should, at least in principle, simplify identifying of shortcomings in the representation of clouds.





Furthermore, the presented validation results show that CCIC can reliably characterize the spatial and temporal distribution of TIWP. This is in contrast to most other available observational TIWP datasets, which in most cases have much more limited spatial and temporal coverage and cannot constrain the full diurnal cycle of TIWP. Therefore, we see great potential for the 540 CCIC data to provide novel insights into cloud processes. Finally, the CCIC data is provided at the same temporal resolution and coverage as many commonly used gridded precipitation products. The combination with precipitation estimates, in particular, may help to better constrain cloud and precipitation processes in models.

## 5.4 Data dissemination

With the conclusion of this validation study, we plan to begin the processing of the CCIC climate data record with the aim 545 of publishing data records for the full temporal extent of the CPCIR and GridSat dataset. However, we will likely not be able to publish vertically-resolved retrieval targets due to storage restrictions. To make these retrievals available for interested researchers, we publish the CCIC retrieval as a Python software package (Amell and Pfreundschuh, 2023). The software package provides a simple command line interface that will allow users to run the retrievals on their own hardware: a high-level description is provided in Appendix D. Information on how to access the latest processed climate data record is available 550 through the software package repository at https://github.com/see-geo/ccic (last accessed: 25 August 2023).

## 6 Conclusions

This study presented and evaluated the CCIC hydrometeor retrieval, which aims to provide a new climate record of ice clouds based on the long record of geostationary IR observations. CCIC leverages novel deep-learning-based retrieval techniques to provide accurate estimates of concentrations of frozen hydrometeors and related cloud properties with spatially and temporally 555 continuous coverage between $70\,^{\circ}$S and $70\,^{\circ}$N. The retrieval has been thoroughly characterized using one year of collocations with CloudSat 2C-ICE and 2B-CLDCLASS measurements and validated against independent measurements of TIWC and TIWP from two flight campaign series and a ground-based cloud radar. Albeit not achieving equally high spatial resolution as the CloudSat reference measurements, the retrieval provides reasonable accuracy in comparison with CloudSat and is consistent with the field-campaign and ground-based radar measurements. Retrieval biases were found to be within $\pm 60\,\%$ a large number 560 of in-situ measurements in both tropical and mid-latitude regimes. The CCIC retrievals are capable of reproducing the diurnal cycle of TIWP as measured by a ground-based radar, which shows that CCIC retrieval ideally complements currently available measurements from CloudSat and the A-train, which are much more limited in the spatial and temporal coverage.

An important question that remains to be answered is the stability of the climate record across the full range of the input datasets, which we aim to address this question in an upcoming study. Nonetheless, the validation experiments presented here 565 cover a fairly long time range outside the time period that has been used for the training of the retrieval, which provides preliminary evidence of the stability of the retrieval. This makes us confident that the CCIC climate record will be a valuable addition to the currently available record of ice cloud properties and hopefully help to improve the understanding of cloud processes in the atmosphere.



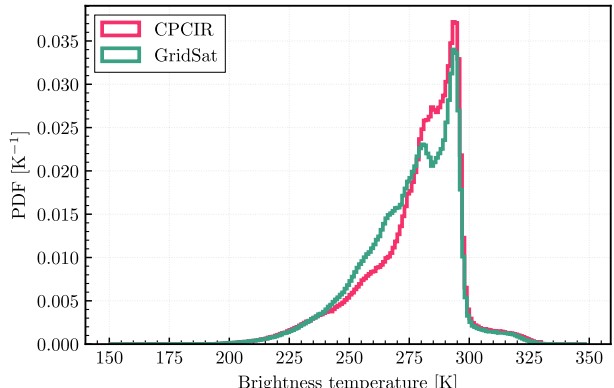

**Figure A1.** Distribution of the brightness temperature closest to 11 µm in the training set.

*Code and data availability.* The code used to produce the results in this paper is publicly available from Amell and Pfreundschuh (2023)
and allows replicating the training database, which is not published due to its size but can be provided upon request. Similarly, all data
sources used in this study are publicly available: CloudSat 2B-CLDCLASS and 2C-ICE products can be downloaded from the CloudSat
Data Processing Center (https://www.cloudsat.cira.colostate.edu, last accessed: 21 August 2023); CPCIR data can be downloaded from
NASA's Goddard Earth Sciences Data and Information Services Center (GES DISC, Janowiak et al., 2017); GridSat can be downloaded
from NOAA's National Climatic Data Center (Knapp and NOAA CDR Program, 2014); HAIC-HIWC data can be downloaded from the
UCAR/NCAR Earth Observing Laboratory data archive (Strapp, 2016b; SAFIRE, 2016; Strapp, 2016a, 2017, 2019; Bennett, 2019); the
OLYMPEX radar observations are available from Heymsfield and Lin (2017) and the in-situ measurements from Poellot et al. (2017); ERA5
can be downloaded from Copernicus Climate Change Service Climate Data Store (CDS, Hersbach et al., 2023); and the Cloudnet data can
be downloaded from the ACTRIS Cloudnet data portal (https://cloudnet.fmi.fi, last accessed: 21 August 2023).

## Appendix A: Training set distributions

The probability distribution functions (PDFs) of brightness temperature, TIWP, and TIWC in the training set are given by
Figs. A1, A2, and A3, respectively. Table A1 shows the frequencies of cloud classes and cloudy reference pixels.

## Appendix B: Training settings

Table B1 shows training choices not mentioned in Sect. 2.1.4. Any parameter not specified corresponds to the default values in
PyTorch 1.13.0 (Paszke et al., 2019a).



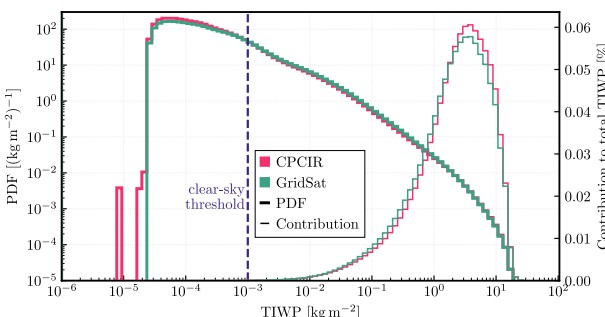

**Figure A2.** TIWP distributions in the training set. Probabilities of TIWP $< 10^{-6}\,\mathrm{kg\,m^{-2}}$ for CPCIR and GridSat: 53.6% and 50.8%, respectively. Clear-sky threshold explained in Sect. 2.1.4.

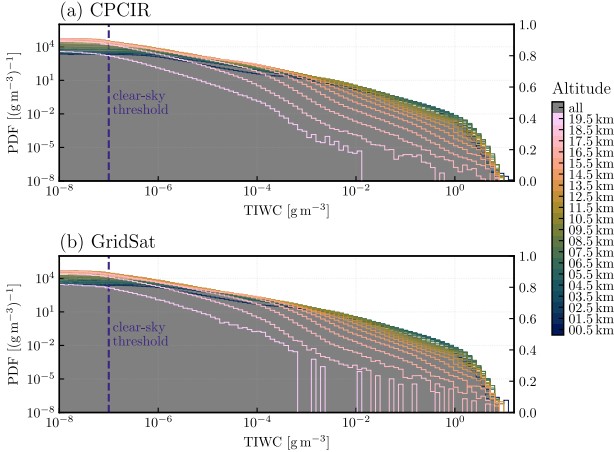

**Figure A3.** TIWC distributions in the training set at the different altitudes. Probabilities of TIWC $< 10^{-8}\,\mathrm{g\,m^{-3}}$ for CPCIR and GridSat for all altitude levels: 84.3% and 83.3%, respectively. Clear-sky threshold explained in Sect. 2.1.4.

## Appendix C: Radar retrievals

In order to compare CCIC retrievals with observations from airborne and ground-based cloud-radar observations in a consistent manner, we have developed a radar-based TIWC retrieval based on the retrievals used in Pfreundschuh et al. (2022b). The input of the retrieval are range-resolved radar reflectivity measurements from which hydrometeor concentrations are derived using the optimal estimation method (OEM, Rodgers, 2000). The retrieval is implemented using the Atmospheric Radiative Transfer Simulator (ARTS, Buehler et al., 2018).





**Table A1.** Cloud class frequencies, in percent, from all levels in the training set.

| Cloud class | GridSat | CPCIR |
|---|---|---|
| No cloud | 91.7 | 92.2 |
| Cirrus | 1.4 | 1.5 |
| Altostratus | 2.2 | 2.0 |
| Altocumulus | 0.6 | 0.6 |
| Stratus | 0.03 | 0.03 |
| Stratocumulus | 0.9 | 0.9 |
| Cumulus | 0.3 | 0.3 |
| Nimbostratus | 2.0 | 1.5 |
| Deep convection | 0.9 | 0.9 |
| Cloudy pixel* | 46.7 | 44.3 |

*According to the 2D cloud mask.

**Table B1.** Training choices not mentioned in text.

| Parameter | Value |
|---|---|
| Batch size | 4 |
| Optimizer | AdamW, learning rate $5 \times 10^{-4}$ |
| Scheduler | Cosine annealing, $T_{\max} = 20$ |

## C1 Forward model

The ARTS-based radar forward model calculates polarized single-scattering radar reflectivities taking into account absorption from gases and particles. Gaseous absorption is modeled in the same way as described in Pfreundschuh et al. (2022b). Particle size distributions of frozen hydrometeors are represented using the single-moment parametrizations by Field et al. (2007).

For liquid hydrometeors the parametrization by Abel and Boutle (2012) is used. The forward model assumes hydrometeors below the freezing level to be liquid and frozen above. The model will thus misrepresent melting particles and liquid particle in convective cores, which will incur and error in the retrieved hydrometeor concentrations.

Particle single-scattering properties are taken from the ARTS single-scattering database (SSDB, Eriksson et al., 2018). Since the assumed shape of ice particle represents a major source of uncertainty for radar-based retrievals of ice concentrations, we

have selected a set of three particles that aim to capture the range of scattering properties in the ARTS SSDB. The particle models used are the 8-Column Aggregate, Large Plate Aggregate and Large Column Aggregate, which also used in the retrievals presented in Pfreundschuh et al. (2022b).

Atmospheric temperature, humidity and liquid cloud water content is taken from ERA5 (Hersbach et al., 2020).



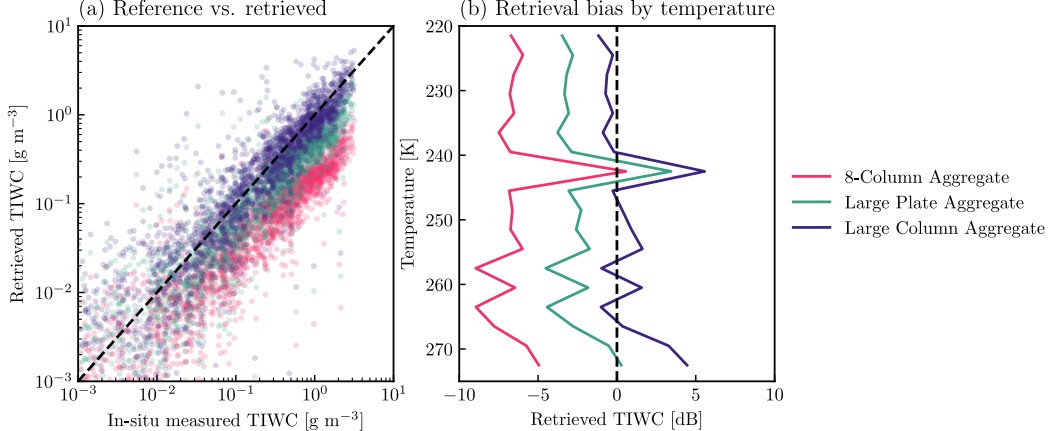

**Figure C1.** Comparison of radar retrievals and in-situ measurements. Panel (a) displays the relation between retrieved TIWC and in-situ measurements collocated at the spatial and temporal resolution of the CPCIR-based CCIC retrievals. Panel (b) displays the mean profiles of the logarithmic ratio of retrieved and in-situ TIWC with respect to the atmospheric temperature.

## C2 HAIC-HIWC

During the Darwin leg of the HAIC-HIWC campaign, the RASTA (Radar Airborne System Tool for Atmosphere) was flown onboard the F20 aircraft of the Service des Avions Français Instrumentés pour la Recherche en Environnement (SAFIRE) research aircraft. The RASTA system comprises multiple radar beams and measures Doppler-spectra of radar reflectivity. For the retrievals used here, only total reflectivity from the zenith and azimuth beams were used in the retrieval. Input observations were resampled to a temporal resolution of 30 s and a vertical resolution of 100 m. The retrievals for the zenith and azimuth

beams are performed independently. The retrievals used the PSD parametrization for tropical regimes by Field et al. (2007).

The radar retrievals from the HAIC-HIWC campaign are evaluated against in-situ measurements in Fig. C1. The in-situ measurements were performed by the same aircraft. The radar-retrieved TIWC at the aircraft position was obtained by linear interpolation between the TIWC retrieved 250 m above and 250 m below the aircraft. The radar retrievals are highly correlated with the in-situ measurements for all particle models. The 8-Column Aggregate and Large Plate Aggregate models underes-

timate the in-situ-measured TWC. The Large Column Aggregate slightly overestimates the in-situ measurements but yields results closest to the in-situ measurements. The mean profiles of the logarithmic ratio of retrieved and in-situ TWC indicate that the Large Column Aggregate, and to lesser degree also the Large Plate Aggregate, yield relatively higher TIWC at lower altitudes.

Table C1 lists the correlation and mean biases of the radar retrieval and the in-situ measurements. The retrieval bias is

largest for the 8-Column Aggregate, which underestimates the mean TIWC by a factor of 4. For the Large Plate Aggregate the mean TIWC is underestimated by a factor of 2. The Large Column Aggregate yields the best agreement with the in-situ measurements.



**Table C1.** Correlation and mean bias for different particle shapes assumed in the radar retrievals from the HAIC-HIWC campaign.

| Particle | Bias [%] | Correlation |
|---|---|---|
| 8-Column Aggregate | -76.63 | 0.92 |
| Large Plate Aggregate | -42.36 | 0.91 |
| Large Column Aggregate | 9.066 | 0.88 |

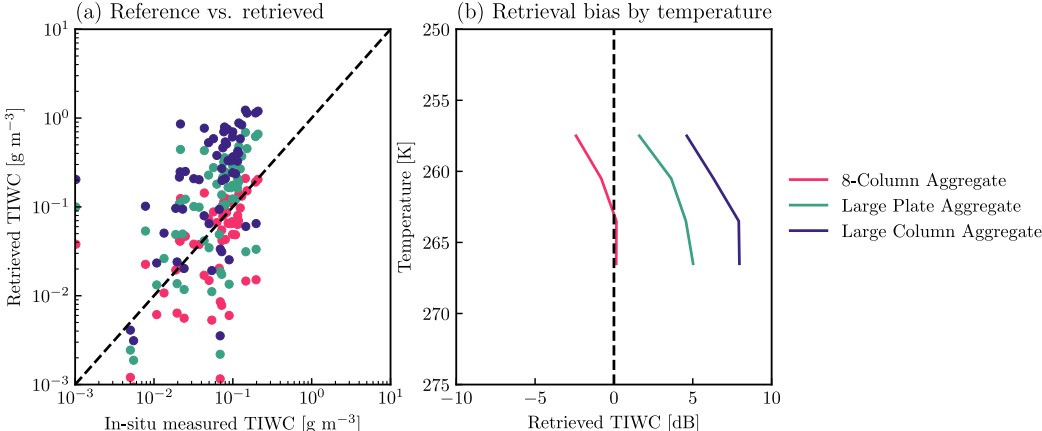

**Figure C2.** Same as Fig. C1 but for the measurements from the OLYMPEX campaign.

## C3  OLYMPEX

During the OLYMPEX campaign, W-band cloud radar observations were conducted by the cloud-radar system (CRS) onboard
the NASA ER-2 aircraft. The retrievals used the PSD parametrization for mid-latitude regimes by Field et al. (2007). Figure C2
shows scatter plots of our CRS TIWC retrievals and collocated in-situ measurements from the UND citation aircraft. To
collocate the radar and in-situ measurements, both were mapped to the spatial and temporal resolution of the CCIC CPCIR
retrieval.

Table C2 lists the mean bias of the radar retrieval and its correlation with the in-situ measurements. The 8-Column Aggregate
yields the best agreement with the in-situ measurements. The Large Plate Aggregate and the Large Column Aggregate particle
models overestimate the in-situ measurements by 118 and 344%, respectively. Despite being lower than for the HAIC-HIWC,
the retrieved TIWC remains well correlated with correlations between 0.73 − 0.74 for all particles.

For the OLYMPEX campaign, the 8-Column Aggregate yields the best agreement with the in-situ measurements. The Large
Plate Aggregate and the Large Column aggregate both overestimate the TIWC. However, compared to the CCIC retrievals,
the Large Plate Aggregate was found to yield biases that were more consistent with the biases between CCIC retrieval and
in-situ measurements. Since most collocations between the CRS radar and the UND in-situ measurements stem from a single
flight, we suspect that the inconsistency of the results in Fig. C2 and the comparison to the CCIC retrievals is due to the



**Table C2.** Correlation and mean bias for different particle shapes assumed in the radar retrievals from the OLYMPEX campaign.

| Particle | Bias [%] | Correlation |
|---|---|---|
| 8-Column Aggregate | -8.1 | 0.74 |
| Large Plate Aggregate | 118.14 | 0.74 |
| Large Column Aggregate | 344.32 | 0.73 |

limited number of collocations of radar retrievals and in-situ measurements and therefore choose the results obtained with the Large-Plate Aggregate as the reference results for the OLYMPEX campaign.

**C4  Cloudnet Palaiseau**

The retrievals for the ground-based W-band cloud radar at the Cloudnet site in Palaiseau use the same retrieval framework as for the airborne radars. Radar reflectivities were averaged over $30\,\mathrm{s}$ and sampled at $4\,\mathrm{min}$ intervals. Retrievals were run for every day of the year 2019.

No in-situ measurements were available to validate the ground-based radar retrievals. However, since retrievals with the
Large Plate Aggregate were found to be consistent with CCIC and in-situ retrievals for the OLYMPEX campaign, we chose the Large Plate Aggregate as the reference particle model.

**Appendix D:  The CCIC software package**

The NN developed and trained for CCIC is publicly available through the CCIC software package (Amell and Pfreundschuh, 2023), which facilitates running the retrievals on any modern computer. Given the fully convolutional neural network nature
of the CCIC NN, retrievals for small regions of interest (ROI) are possible. The retrieval for large areas is implemented by dividing the input image in tiles that partly overlap, running the retrieval for each of these tiled areas, and finally aggregating the retrievals with a weighted average, with weights inversely proportional to the distance to the center of each tile. This implementation aims to reduce the memory footprint of the inference and minimizes any CNN edge effects.

The software enables the user to choose, among others, a ROI, the variables of interest, and a CI complementing the TIWP
expected value. The infamous quantile crossing problem (lack of monoticity in the QPD) was detected in the retrieval of continuous variables, but, as discussed in Appendix E, it was deemed negligible for the default setting of a 90% CI. Input data with invalid pixels is supported, since the CCIC NN likely leverages contextual information to provide a retrieval. Finally, the CCIC software package saves the retrievals either as compressed netCDF files or as Zarr files. The latter file format facilitates distributed access as well as significantly reduces the file size through a custom compression algorithm. Regardless of the file
format used, the CF conventions (Eaton et al., 2022) are followed.




## Appendix E: Is quantile crossing a problem for CCIC?

One of the properties that any cumulative distribution function (CDF) must have is to be a right-continuous monotone increasing function. With the formulation of QRNNs used, there is the possibility that the QPDs obtained violate this statistical property; for example, the quantile $x_{\tau_i}$ at level $\tau_i$ being larger than the quantile $x_{\tau_{i+1}}$ at level $\tau_{i+1} > \tau_i$, yet this would imply that $\tau_{i+1} <$ $\tau_i$ (cf. $\mathrm{P}(X \leq x_\tau) = \tau$). This is an infamous problem referred to as quantile crossing, and arises from quantile regression itself. Depending on the formulation of the problem, it cannot be avoided: a minimal example is doing linear quantile regression for two levels for a cloud of points in $\mathbb{R}^2$. In this case, the lines obtained will extremely likely cross on the real line. Therefore, quantile crossing can require additional analysis or caution when drawing conclusions.

Inference with the GridSat training data for June 2006 is used here to analyze the presence and implications of quantile crossing for the retrievals offered by the CCIC software package, using a 90% CI. The Spearman correlation coefficient $\rho_S \in$ $[-1, +1]$ measures the monotonic relationship between two variables, with $\pm 1$ indicating a perfect monotonic relationship. Consequently, it can be used to assess the deviation of a QPD from a monotone function, that is, the presence and magnitude of quantile crossing by computing $\rho_S$ between the estimated quantiles and the quantile levels. This correlation results $\rho_S \geq 99.8\%$ for TIWP and $\rho_S \geq 80.6\%$ for TIWC for all retrievals; these results are consistent with the argument that the retrieval of TIWC is more difficult. These high $\rho_S$ values indicate that the QPDs are generally close to being perfect monotonic functions, but not all of them. However, these last QPDs can be considered to be relatively rare, as the median of the distributions of $\rho_S$ for any of the three variables is practically $1$.

There are numerous approaches to address quantile crossing. An option is to process the QPD with a linear isotonic regression of the quantiles $x_\tau$ by solving

$$
\begin{aligned}
\text{minimize} \quad & \sum_{\tau_i \in \mathcal{T}} \left( x_{\tau_i}^{(c)} - x_{\tau_i} \right)^2 \\
\text{subject to} \quad & x_{\tau_i}^{(c)} \leq x_{\tau_j}^{(c)} \quad \forall \tau_i \leq \tau_j
\end{aligned}
\tag{E1}
$$

and obtain quantiles $x_\tau^{(c)}$ that are monotonic. The idea behind this approach is to have a simple, computationally-friendly method that respects as much as possible the QPD given by the network. Figure E1 shows the QPDs with the worst $\rho_S$ for the `tiwp` and `tiwc` variables, as well as the corrected QPDs (CQPDs) through a linear isotonic regression. At least three things can be spotted in Fig. E1: the highly non-linear shape of the QPDs, the quantile levels with crossing quantiles, and that the QPD and CPQD expected values result virtually identical. Intuitively, nearby quantile levels have higher odds of experiencing quantile crossing than distant levels. Consequently, a large CI is more robust: all quantiles defining the 90% CI for the retrievals analyzed do not cross.

Computing the signed relative percent difference between a scalar $x$ derived from the QPD and the analogous value derived from the CQPD, referred as $x^{(c)}$, enables comparison among different orders of magnitude. In this analysis their mean is used as a reference, that is

$$
d_1(x, x^{(c)}) = \frac{x - x^{(c)}}{(|x| + |x^{(c)}|)/2}
\tag{E2}
$$



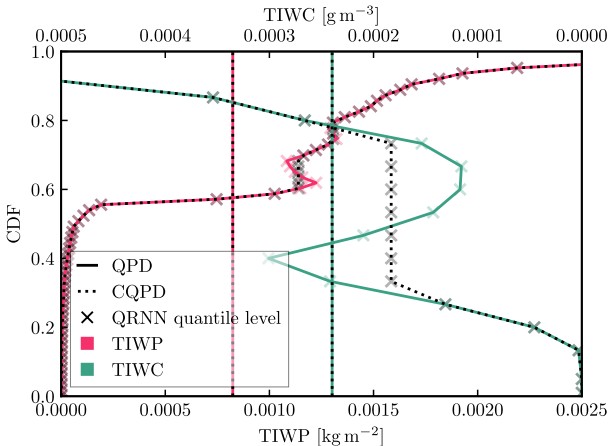

**Figure E1.** QPDs with the worst $\rho_S$ (99.8% for TIWP and 80.6% for TIWC), where the vertical lines indicate the expected value of the corresponding distribution. Note the different abscissa axes and that the plot limits were deliberately selected to draw attention to the crossing quantiles.

with $d_1(0,0) = 0$, and therefore ranges from $-2$ to $+2$. The distribution of $d_1$ values for all TIWP and TIWC variables resembles a delta function, with 99% of the values being zero, and the quantiles defining the 99.9% equal-tailed distribution for TIWP-derived $\rho_S$ being -0.0011 and 0.0151. Consequently, there may be instances of quantile crossing, but its impact on 695 the variables reported by the CCIC software package is negligible when contrasted with enforcing monotonicity of the QPDs through linear isotonic regression.

*Author contributions.* All authors contributed to the project through discussions and feedback. AA and SP collected the data, developed the CCIC software package, performed the data analysis, and prepared the manuscript. SP implemented the neural network and monitored its training, with feedback from AA, and executed the radar retrievals. PE provided scientific advice and enriched the manuscript.

*Competing interests.* The authors declare that they have no conflict of interest.

*Acknowledgements.* The contributions from SP and PE were covered by Swedish National Space Agency (SNSA) grant 154/19. The computations were performed on resources at Chalmers Centre for Computational Science and Engineering (C3SE) provided by the National Academic Infrastructure for Supercomputing in Sweden (NAISS) and supported by Chalmers AI Research Centre (CHAIR). We also acknowledge the following data actors: CloudSat Data Processing Center for the 2B-CLDCLASS and 2C-ICE data products; NASA GES DISC 705 for the CPCIR data; NOAA's National Climatic Data Center for the GridSat data as well Ken Knapp and colleagues for developing GridSat



for NOAA'S Climate Data Record program; the flight crew and scientists involved in the HAIC-HIWC campaigns as well as UCAR/NCAR Earth Observing Laboratory data archive for providing the corresponding datasets; Julien Delanoë for the RASTA radar data from the Darwin flights of the HAIC-HIWC campaigns and the BASTA data from the Cloudnet site in Palaiseau; the flight crew and scientists involved in the collection of the CRS radar measurements onboard NASA ER-2 aircraft and the Nevzorov probe measurements onboard the UND Citation aircraft; Copernicus Climate Change Service CDS for the ERA5 data; and ACTRIS as well as the Finnish Meteorological Institute for the Cloudnet data.

The computations for this study were performed using several freely available programming languages and software packages, most prominently the Python language (The Python Language Foundation, 2018), the IPython computing environment (Perez and Granger, 2007), the NumPy package for numerical computing (van der Walt et al., 2011), Xarray (Hoyer and Hamman, 2017), PyTorch (Paszke et al., 2019b), Satpy (Raspaud et al., 2022) for the processing of satellite data, and Matplotlib (Hunter, 2007) as well as cartopy (Met Office, 2010 - 2015) for generating figures.



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
