# Peer review of "The Chalmers Cloud Ice Climatology: Retrieval implementation and validation"

_EGUsphere, 2023_

## Referee Comment (RC2)

**1  General Comments**

Overall, this is a very well-written and impressive paper. Although the CCIC algorithm is a proof of concept, the results are pretty promising. For instance, this method is much more advanced and skillful than the brightness temperature-threshold approach to track deep convection.

The CCIC algorithm shows good skill in retrieving ice water path (IWP), and therefore, there is a good potential to create usable long-term day and night IWP datasets, that could help constrain clouds in climate models. Somewhat surprisingly, needing only 11 micron channel as input, the QRNN-based method shows some skill in retrieving vertically resolved ice water content. However, I expect the 3D retrieval to be very uncertain in unusual atmospheric situations. This comment pertains to the stability of the retrieval, which the authors indicate will be assessed in the continuation work.

For me, this is already a good paper and could probably be published as is, but I have a few comments on elements of the paper that should be clarified.

**1.1  Specific comments**

**input data**

- Since the input data is only tested on geostationary data with no input from polar-orbiting satellites, it is worth mentioning that high latitudes are not represented in the study/or something about the likely difficulties in retrieving IWP over snow-covered surfaces. This fact is pertinent since, as far as I understand, GridSat (or at least the new ISCCP-NG, another similar global geostationary dataset) may include polar-orbiting satellites to fill in the missing data at the poles in the future.

- The datasets apply inter-satellite normalization. ” This is not obvious. One method of ”normalizing” the geostationary satellites is to use spectral band adjustments to make all the satellite's 11-micron channels look like a particular sensor, for instance, the SEVIRI 11-micron channel. Is this how it was done? Either way, more information is needed here.

**training data**

- The existence of the 2C-ICE equivalent dataset, DARDAR, should be mentioned somewhere, at least in reference, and possibly half a sentence on why 2C-ICE was chosen as the reference dataset here.

- The authors rightfully point out that the largest source of uncertainties in IWP retrievals is the assumed ice particle microphysical model. However, nothing is mentioned about which microphysical model the 2C-ICE IWP retrievals assume. This needs to be mentioned, especially as it is rightfully considered when retrieving IWP from ground-based Radar.

**validation**

- Cloudnet offers several years of W-band data and more sites than just the one, Palaiseau, in France. Why (only) this site? For instance, I don't know if it is too far North, but Norunda in Sweden would add sub-arctic conditions to the validation. A comment would suffice here.

---

## Author Comment (AC1)

**Response to RC1 on egusphere-2023-1953**

The Chalmers Cloud Ice Climatology: Retrieval implementation and validation
Preprint `https://doi.org/10.5194/egusphere-2023-1953`

Adrià Amell        Simon Pfreundschuh*        Patrick Eriksson

April 24, 2024

*Text from the Anonymous Referee is presented in* gray *and ours in black.*

We would like to thank the reviewer for their thorough review of our work and the constructive comments. We are convinced that the comments helped us to improve our manuscript. We hope that we could successful address the reviewer's concerns.

**Changes to the manuscript**

While the manuscript was in review we discovered a mistake in the radar retrievals from the Palaiseau cloud radar, which used a tropical instead of a mid-latitude PSD parametrization. For the revised manuscript, we have updated the results of the ground-based TIWP retrievals. This did not change the results considerably.

In addition to the issues highlighted by the reviewers, we have also corrected a number of smaller mistakes in the figures included in the manuscript.

**Major comments**

1. While the paper is well-written and includes multiple statistical examples demonstrating the efficacy of the machine learning technique, it lacks maps and curtain plots illustrating the geographic representation of CCIC retrievals. To convincingly demonstrate the representativeness of these estimates, such examples are essential. The authors could add for example:

   - Monthly global IWP maps showing the CCIC against the cloudsat estimates (no need to subsample the CCIC, just show that the global distribution is as expected)
   - Percentage difference in these types of maps.
   - A latitude – altitude cross section of cloud ice fraction, again comparing versus the cloudsat one.
   - global maps of IWC at different levels showcasing the variation with height
* * *
*simon.pfreundschuh@colostate.edu

- Cross sections of IWC through different longitudes

We agree with the reviewer that the initial manuscript provided insufficient evidence of CCIC's capability to capture the spatial distribution of TIWP and TIWC.

To address this, we will extend our analysis of the CCIC retrieval results on the test dataset, which comprises one full year of collocated geostationary observations and corresponding CloudSat measurements. Due to the sparse sampling of the CloudSat observations, we have decided against assessing the spatial distributions of the retrieval results and errors by month as the results would be extremely noisy. We will extend the manuscript with the figures shown in Fig. RC1.1 and Fig. RC1.2, which display the distribution of retrieved and reference TIWP and the zonal means and biases of the TIWC retrieval, respectively.

[Figure]

figs/global_dist_tiwp.pdf

Figure RC1.1: Spatial distribution of retrieved TIWP and 2C-ICE-based reference TIWP for the CPCIR and GridSat test datasets over the domain covered by CCIC. Panels (a) and (b) show the retrieved TIWP aggregated to a resolution of 5°. Panels (c) and (d) show the corresponding distributions of the reference TIWP measurements. Panels (e) and (f) show the biases relative.

[Figure]

Figure RC1.2: Zonally averaged distributions of retrieved TIWC and 2C-ICE-based reference TIWC for the CPCIR and GridSat test datasets over the domain covered by CCIC. Panels (a) and (b) show the retrieved and reference TIWC for the CPCIR observations aggregated to a resolution of 2.5°. Panel (c) show the truncated relative bias of the retrievals. Panels (d), (e), and (f) show the corresponding distributions of the GridSat-based retrieval.

We also provide maps of retrieved and reference TIWC at discrete altitude levels and zonal averages of retrieved and reference TIWC for different longitude bands in Fig. RC1.3 to Fig. RC1.7. While these results provide further evidence of the ability of CCIC to capture the three-dimensional distribution of TIWC in the atmosphere, we do not plan to include them in the manuscript as it already contains a large number of figures.

[Figure]

Figure RC1.3: Distribution of retrieved and 2C-ICE-based reference TIWC measurements for CCIC CPCIR retrievals at three atmospheric levels. Column 1 displays the mean retrieved TIWC. Column 2 displays the corresponding reference TIWC. Column 3 displays the truncated relative error. Rows contain the results for the atmospheric levels at altitudes of 5.5, 10.5 and 15.5 km, respectively.

[Figure]

Figure RC1.4: As Fig. RC1.3 but for GridSat-based CCIC retrievals.

Longitudes: $[-180°, -160°]$

[Figure]

Figure RC1.5: As Fig. RC1.2 but for longitudes within $-180°$ and $-160°$.

Longitudes: $[-40°, -20°]$

[Figure]

Figure RC1.6: As Fig. RC1.2 but for longitudes within $-40°$ and $-20°$.

[Figure]

Figure RC1.7: As Fig. RC1.2 but for longitudes within 20° and 40°.

**Specific comments**

1. The title is not representative of the context of the manuscript. There is no mention of constructing a climatology or anything of that sort.

   We thank the reviewer for pointing out that the introduction failed to properly present the CCIC project and the scope of the manuscript. We have reformulated the abstract to state that CCIC is a novel cloud-property datasets. Moreover, we have reformulated the two paragraphs starting in l. 65 of the revised manuscript to clearly state that the presented manuscript describes the first step towards the production of an ice water path climate record using the presented retrieval.

2. Line 41: "For the study of processes on annual and decadal scales it is therefore necessary to find ways to make better use of observations with a long record of availability". The authors should mention that several IWP records exist with annual and even decadal scales, such as the ones from MODIS, Aura MLS, Odin SMR, CloudSat, etc. As currently written, the introduction implies that such records do not exist.

   We thank the reviewer for pointing out this shortcoming of our manuscript. We will extend the introduction with a list of currently available records of comparable TIWP estimates and their respective shortcomings.

3. Further since CCIC provides IWC, the authors could compare partial IWP versus those records matching their respective altitude coverage. The comparison versus the campaigns is limited to a few periods and it is limited geographically.

   As per the reviewer's suggestion, we will extended the analysis of the CCIC retrieval results to provide a more detailed assessment of the three-dimensional distribution or retrieved TIWC, as detailed in the response to major comment 1. The analysis is based on a full year of collocations with 2C-ICE estimates and thus covers the full geographical extent of the CCIC retrievals.

   While the comment seems to suggest to also extend our comparison to estimates from limb-sounding instruments such as Aura MLS or Odin SMR, we choose not to do this as we do not think that this would offer any benefit over the comparison against the CloudSat/CALIPSO-based estimates.

   Finally, we would like to point out that, while the individual field campaigns used in the validation are naturally limited in their spatial and temporal coverage, they cover both tropical and mid-latitude climate regimes and extend from instantaneous to seasonal time scales. With this, they exceed the scope of most validation efforts of TIWP/TIWC products that were able to find in published literature (Deng et al., 2010, 2013; Eriksson et al., 2008; Wu et al., 2008; Barker et al., 2008).

4. Line 17: "considerable skill" is a qualitative description please provide a more quantitative description.

   We will reformulate the sentence in question to provide quantitative accuracy estimate derived from independent test data.

5. Line 20: "first order" is a qualitative description please provide a more quantitative description.

   We will remove the formulation 'first order' and instead list the linear correlation coefficient and bias of the TIWC estimates in the previous paragraph.

6. Line 45: please describe the rationale behind only using the 11micron channel. Presumably additional channels could provide more information.

   We chose the 11 μm channel because it provides the best temporal and spatial coverage throughout the available record of geostationary satellite observations. While the GridSat B1 product also includes visible and water vapor imagery, which could likely help to improve the retrieval, they are not always available and therefore not considered in the current CCIC retrievals. We will revise the paragraph in question to include the motivation for this design choice.

7. Line 55: "Estimates of TIWP differ widely between". Please give the ranges, this would allow you to later show how well (or bad) CCIC estimates are.

   We will rewrite the relevant parts of the introduction and include the ranges of disagreement between satellite-based IWP estimates.

8. Line 87: why not use lat lon info as well? And day of the year?

   We do not include spatial and/or seasonal context in the retrieval input data as we want the underlying neural network to learn relations between the satellite observations and the corresponding cloud properties and not their variability with respect to geographical location and season. Since the training data period is limited from mid-2006 through 2009, including geographical coordinates and seasonal information could limit the retrieval's ability to reproduct changes in the regional or seasonal variability of clouds outside of the training data period.

9. Line 108: The use of "2D" here is confusing since the authors are talking about profiles, I suggest deleting it.

   We will replace '2D' with 'horizontal'.

10. Line 113 – Line 119: A schematic of this entire procedure will be appreciated. Also, what is the treatment for the uncertainties in 2C-ICE

    We acknowledge the importance of making our training-data-generation process transparent and reproducible. To of clouds this end, we have published all relevant code in the repository accompanying this manuscript. However, since the manuscript already contains a large number of figures, we chose not to include a schematic of the data extraction process as we consider it of minor interest for the general audience.

    We will add a sentence referring the interested reader to the relevant code to the revised manuscript.

    Regarding the uncertainties of the 2C-ICE product: We treat the 2C-ICE estimates as ground-truth and do not make any effort to model the associated uncertainties. Synetergistic radar-lidar retrievals of ice hydrometeor

concentrations have to be regarded as the most most accurate global measurements of TIWC and TIWP. Although even these combined radar-lidar estimates remain affected by significant systematic uncertainties, largely due to the underlying microphysical assumptions on particle shape and distribution, these uncertainties are not well characterized due to the limited amount of work that compares them with in-situ measurements (Deng et al. (2010, 2013) are the only studies that we are aware of). Since what ultimately matters to future users of CCIC is the uncertainty in the estimates provided by CCIC, we chose to extensively validate the resulting CCIC estimates instead of trying to handle uncertainties in the 2C-ICE product upfront.

11. Line 136: "Training scenes of 384×384 pixels". Is the geographical size of this scene important? Is there an impact for using smaller or bigger scenes? Why this particular size.

    We chose this size since it allows use to extract randomly rotated crops of size 256×256 pixels without generating invalid values. The scene size of 256×256, which is ultimately used in the retrieval, was chosen because it corresponds to scenes covering more than 900 km in zonal and meridional extent. The resulting scenes should thus contain information on the mesoscale and, to limited extent, synoptic-scale context of the retrieval.

    To make this point clear, we have reformulated the paragraph in question to provide a better description of the training-dataset generation and the underlying motivation.

12. Line 136: "The process involved randomly selecting a pixel with valid reference data as the starting point and then adding a random zonal offset." I don't really understand this please clarify

    We have reformulated the paragraph in question to describe the training-data generation process more clearly.

13. Line 140: lower → coarser

    We will adopt this suggestion in the revised version of the manuscript.

14. Figure 2 caption: Why is the brightness temperature normalized?

    Normalizing the inputs to neural networks is common practice and generally leads to better results and faster training convergence.

15. Line 155: Good is the enemy of great. The authors should explore tuning of these parameters or rephrase this sentence to state that minimal tuning was required.

    It is generally acknowledged that exhaustive tuning of all architecture-related hyperparameters of a neural network model is too resource consuming to be practically feasible and principled architecture search remains and activate area of machine-learning research (Ren et al., 2021). We therefore consider the reviewer's request out of scope for our work.

16. Figure 3, and 4 captions: specify the locations. for example, Aug 2015, and Jul-Aug 2018 flights were over the US and the US nearest oceans. Or,

Flights took place over the Olympic Peninsula in the Pacific Northwest of the United States.

We will add the requested flight locations to the captions of Fig. 3 and 4.

17. Line 229: Which of the four periods is the Darwin campaign?

We will extend the sentence in question to state that the Darwin campaign is the first set of flights of the HAIC-HIWC campaign.

18. Line 237: Specify frequency

Following the reviewer's suggestion, we have replaced 'W-band' by '94-GHz'.

19. Line 247: Which year?

The year was 2019. We will include this information in the revised manuscript.

20. Figure 5: panels g and h should say Cloud classification and cloud classification (retrieved) respectively.

We will update the panel titles in the revised manuscript.

21. Figure 6 caption should mention cloudsat somewhere, as well as the period use for this comparison.

We will add the requested information to the figure caption.

22. Section 3.2.1. It is not clear which period this comparison cover.

To make it clear to the reader that all results in this sub-section are derived using the independent test data, we add an introductory sentence to Section 3.2. Furthermore, we now state that the distributions were computed from the test dataset.

23. Line 297: This should be shown as a separate subsection to emphasize its importacnce: Zero order comparison of IWP and IWC (for example)

Following the reviewers suggestion, we have added a new section titled 'Consistency of retrieved TIWP and TIWC profiles' and moved the discussion of the consistency of the retrieved TIWP and TIWC there.

24. I think the whole classification is barely working and the authors should just not show any of those results.

We respectfully disagree with the reviewer's comment but acknowledge that results presented in the first version of the manuscript may give an overly negative view of the retrieval's capabilities. We would like to point out, however, that the classification results from the case study presented in Fig. 5 demonstrate the ability of the retrieval to distinguish the types of the principal cloud systems in the scene. To provide a more comprehensive picture of the retrieval's classification skill, we will add curtain plots (conceptually similar to the ones the reviewer requested to demonstrate the skill of TIWC retrieval) showing the spatial distributions of the retrieved and reference cloud classes with respect to latitude and altitude. The results, shown in Fig. RC1.8 below, show good agreement between the distributions of the retrieved and reference cloud classes for

all cloud classes except the stratus (St) class, whose frequency in the training dataset is only 0.03 %. We consider this compelling evidence that the CCIC retrieval, in fact, has skill in classifying different cloud types.

[Figure]

Figure RC1.8: Spatial distribution of retrieved cloud classes and 2B-CLDCLASS-based reference cloud classes for the test samples from the year 2010. Each row of panels shows the distribution of one of the 8 cloud classes distinguished by the 2B-CLDCLASS product. The first column shows the results retrieved from CPCIR observations while the second column shows the corresponding reference distribution. Column three and four show the corresponding results for retrievals based on GridSat observations.

In addition to adding the curtain plots to the manuscript, we added text that discusses the results and points out that the confusion matrix shown in Fig. 10 assess the classification of cloud layers at a vertical resolution of 1 km, which leads to high uncertainties in the classification of individual layers.

25. Figure 13 is missing the conditional mean line

We have made the conscious decision to not show conditional mean lines

for the validation results as some of the campaigns have very few samples causing the conditional mean lines to become very noisy. While the HAIC-HIWC in-situ data used in Fig. 13 contains sufficient samples to show the conditional mean lines (see Fig. RC1.9 below) we feel that we would have to add conditional mean lines to all following scatter plots to be consistent. Since we do not think that the conditional mean lines add significant information to the scatter plots, we have decided against showing conditional mean lines for the validation results.

[Figure]

Figure RC1.9: As in Fig. 13 from the preprint, but with conditional mean lines.

26. Line 417: CCIC is not representing the diurnal variability well, it is really flat.

   While the CCIC results certainly do not represent the diurnal variability perfectly, the retrieved and reference diurnal cycles show a high degree of correlation. To make this point clear we will add a table containing the relative biases and linear correlation coefficients of the retrieved mean TIWP to the manuscript. These results show that the correlation of the diurnal cycles calculated over the full year is is 0.86 (0.97) for CPCIR-based (GridSat-based) retrievals and does not fall below 0.74 for any of the assessed three-month periods.

27. Line 438: provide an estimate of the uncertainties associated with the estimates of the ice hydrometeors.

Table RC1.9: Relative bias and linear correlation coefficient of the diurnal cycles retrieved using CCIC compared to those derived from ground-based cloud-radar observations.

| | CPCIR | | GridSat | |
|---|---|---|---|---|
| Time period | Bias [%] | Correlation coeff. | Bias [%] | Correlation coeff. |
| All year | 27.43 | 0.86 | 34.59 | 0.97 |
| DJF | 37.97 | 0.92 | 23.58 | 0.97 |
| MAM | 53.04 | 0.81 | 73.47 | 0.92 |
| JJA | 45.76 | 0.74 | 30.92 | 0.96 |
| SON | -0.66 | 0.84 | 25.79 | 0.92 |

We have extended the sentence in question to reference the validation study of A-train-based IWC retrievals by Deng et al. (2013) and mention the biases of up to 59 % compared to in-situ measurements.

28. Line 456: "Despite these encouraging results, CCIC should still be considered a proof of concept. CCIC's principal objective remains to explore the potential of modern deep-learning techniques to expand the observational climate record of ice clouds". This should be mentioned upfront in the abstract and the introduction.

Since we have in the mean time processed the full observational record of available geostationary observations, it is not adequate anymore to consider CCIC merely a proof of concept. We will therefore the sentence from the revised manuscript.

29. Table A1: "Cloudy pixel" Cloudsat or retrieved? how come the cloudy pixel is 40ish while the no cloud is 97 %, please clarify

"Cloudy pixel" refers to cloudy profiles containing a cloud anywhere at the 20 vertical levels used by CCIC. The 97 %, on the other, hand refer to the fraction of non-cloudy levels. To make this point clearer we will replace 'cloudy pixel' with 'cloudy profile' in Table 1 and update the caption.

**References**

H. W. Barker, A. V. Korolev, D. R. Hudak, J. W. Strapp, K. B. Strawbridge, and M. Wolde. A comparison between cloudsat and aircraft data for a multilayer, mixed phase cloud system during the canadian cloudsat-calipso validation project. *Journal of Geophysical Research: Atmospheres*, 113(D8), 2008. doi: https://doi.org/10.1029/2008JD009971. URL https://agupubs.onlinelibrary.wiley.com/doi/abs/10.1029/2008JD009971.

M. Deng, G. G. Mace, Z. Wang, and H. Okamoto. Tropical composition, cloud and climate coupling experiment validation for cirrus cloud profiling retrieval using cloudsat radar and calipso lidar. *Journal of Geophysical Research: Atmospheres*, 115(D10), 2010. doi: 10.1029/2009JD013104.

M. Deng, G. G. Mace, Z. Wang, and R. P. Lawson. Evaluation of several a-train ice cloud retrieval products with in situ measurements collected

during the sparticus campaign. *Journal of Applied Meteorology and Climatology*, 52(4):1014 – 1030, 2013. doi: 10.1175/JAMC-D-12-054.1. URL https://journals.ametsoc.org/view/journals/apme/52/4/jamc-d-12-054.1.xml.

P. Eriksson, M. Ekström, B. Rydberg, D. L. Wu, R. T. Austin, and D. P. Murtagh. Comparison between early odin-smr, aura mls and cloudsat retrievals of cloud ice mass in the upper tropical troposphere. *Atmospheric Chemistry and Physics*, 8(7):1937–1948, 2008. doi: 10.5194/acp-8-1937-2008. URL https://acp.copernicus.org/articles/8/1937/2008/.

P. Ren, Y. Xiao, X. Chang, P.-y. Huang, Z. Li, X. Chen, and X. Wang. A comprehensive survey of neural architecture search: Challenges and solutions. *ACM Comput. Surv.*, 54(4), may 2021. ISSN 0360-0300. doi: 10.1145/3447582. URL https://doi.org/10.1145/3447582.

D. L. Wu, J. H. Jiang, W. G. Read, R. T. Austin, C. P. Davis, A. Lambert, G. L. Stephens, D. G. Vane, and J. W. Waters. Validation of the aura mls cloud ice water content measurements. *Journal of Geophysical Research: Atmospheres*, 113(D15), 2008. doi: https://doi.org/10.1029/2007JD008931. URL https://agupubs.onlinelibrary.wiley.com/doi/abs/10.1029/2007JD008931.

---

## Author Comment (AC3)

**Response to RC2 on egusphere-2023-1953**

The Chalmers Cloud Ice Climatology: Retrieval implementation and validation
Preprint `https://doi.org/10.5194/egusphere-2023-1953`

Adrià Amell          Simon Pfreundschuh[*]          Patrick Eriksson

April 24, 2024

*Text from the Anonymous Referee is presented in gray and ours in black.*

We thank the reviewer for the nice and concise summary of the paper and for the overall review. We are convinced that the comments helped us to improve our manuscript. As the reviewer noticed, the stability of the retrieval will be assessed with a follow-up work. We address the specific comments below.

**Changes to the manuscript**

While the manuscript was in review we discovered a mistake in the radar retrievals from the Palaiseau cloud radar, which used a tropical instead of a mid-latitude PSD parametrization. For the revised manuscript, we have updated the results of the ground-based TIWP retrievals. This did not change the results considerably. We have also corrected a number of smaller mistakes in the figures included in the manuscript.

**Specific comments**

**input data**

- Since the input data is only tested on geostationary data with no input from polar-orbiting satellites, it is worth mentioning that high latitudes are not represented in the study/or something about the likely difficulties in retrieving IWP over snow-covered surfaces. This fact is pertinent since, as far as I understand, GridSat (or at least the new ISCCP-NG, another similar global geostationary dataset) may include polar-orbiting satellites to fill in the missing data at the poles in the future.

  As suggested by the reviewer, we will add a remark regarding the limited latitudinal coverage of the CCIC retrievals to the paragraph starting in line 65 of the revised manuscript.

  Moreover we will add a paragraph of to the discussion secion in which we discuss the prospects of applying the CCIC retrieval to high-latitude and polar regions.
* * *
[*]simon.pfreundschuh@colostate.edu

- The datasets apply inter-satellite normalization. " This is not obvious. One method of "normalizing" the geostationary satellites is to use spectra band adjustments to make all the satellite's 11-micron channels look like a particular sensor, for instance, the SEVIRI 11-micron channel. Is this how it was done? Either way, more information is needed here

While we do not apply any normalization ourselves, we used the GridSat and CPCIR data as is, which have been already normalized. We updated the text in question to provide a high-level summary of the normalization applied by these datasets.

**training data**

- The existence of the 2C-ICE equivalent dataset, DARDAR, should be mentioned somewhere, at least in reference, and possibly half a sentence on why 2C-ICE was chosen as the reference dataset here.

We will add this information to the revised manuscript.

- The authors rightfully point out that the largest source of uncertainties in IWP retrievals is the assumed ice particle microphysical model. However, nothing is mentioned about which microphysical model the 2C-ICE IWP retrievals assume. This needs to be mentioned, especially as it is rightfully considered when retrieving IWP from ground-based Radar.

We will add the information regarding the particle habit and PSD used by 2C-ICE to the discussion secion.

**validation**

- Cloudnet offers several years of W-band data and more sites than just the one, Palaiseau, in France. Why (only) this site? For instance, I don't know if it is too far North, but Norunda in Sweden would add sub-arctic conditions to the validation. A comment would suffice here

The Norunda radar at 60.0860°, unfortunately, is right oustide the latitude covered by the CCIC CPCIR retrievals. We only looked into latitudes covered by both datasets. From these, the radar in Palaiseau can be considered to be the Cloudnet site with the most complete and high quality W-band radar data record, in particular for 2019, the year used (2023 is complete as well, but it is when the manuscript was written). Furthermore, we did not want to overload the paper with figures and restricted the validation to one radar. An additional reason for this is that running the radar-only retrievals for a full year computationally expensive. Hence the choice of Palaiseau.

We will add a sentence summarizing this motivation to Sect. 2.2.3.

---

## Author Response (AR1)

**Response to RC1 on egusphere-2023-1953**

The Chalmers Cloud Ice Climatology: Retrieval implementation and validation
Preprint `https://doi.org/10.5194/egusphere-2023-1953`

Adrià Amell          Simon Pfreundschuh*          Patrick Eriksson

April 24, 2024

*Comments from the Anonymous Referee are presented in gray and our responses in black. Line numbers are given with respect to the revised manuscript.*

We would like to thank the reviewer for their thorough review of our work and the constructive comments. We are convinced that the comments helped us to improve our manuscript. We hope that we could successful address the reviewer's concerns.

**Changes to the manuscript**

While the manuscript was in review we discovered a mistake in the radar retrievals from the Palaiseau cloud radar, which used a tropical instead of a mid-latitude PSD parametrization. For the revised manuscript, we have updated the results of the ground-based TIWP retrievals. This did not change the results considerably.

In addition to the issues highlighted by the reviewers, we have also corrected a number of smaller mistakes in the figures included in the manuscript.

**1 Major comments**

**Comment 1**

While the paper is well-written and includes multiple statistical examples demonstrating the efficacy of the machine learning technique, it lacks maps and curtain plots illustrating the geographic representation of CCIC retrievals. To convincingly demonstrate the representativeness of these estimates, such examples are essential. The authors could add for example:

- Monthly global IWP maps showing the CCIC against the cloudsat estimates (no need to subsample the CCIC, just show that the global distribution is as expected)

- Percentage difference in these types of maps.
* * *
*simon.pfreundschuh@colostate.edu

- A latitude – altitude cross section of cloud ice fraction, again comparing versus the cloudsat one.

- global maps of IWC at different levels showcasing the variation with height

- Cross sections of IWC through different longitudes

**Author response**

We agree with the reviewer that the initial manuscript provided insufficient evidence of CCIC's capability to capture the spatial distribution of TIWP and TIWC.

To address this, we will extend our analysis of the CCIC retrieval results on the test dataset, which comprises one full year of collocated geostationary observations and corresponding CloudSat measurements. Due to the sparse sampling of the CloudSat observations, we have decided against assessing the spatial distributions of the retrieval results and errors by month as the results would be extremely noisy. We will extend the manuscript with the figures shown in Fig. RC1.1 and Fig. RC1.2, which display the distribution of retrieved and reference TIWP and the zonal means and biases of the TIWC retrieval, respectively.

[Figure]

Figure RC1.1: Spatial distribution of retrieved TIWP and 2C-ICE-based reference TIWP for the CPCIR and GridSat test datasets over the domain covered by CCIC. Panels (a) and (b) show the retrieved TIWP aggregated to a resolution of 5°. Panels (c) and (d) show the corresponding distributions of the reference TIWP measurements. Panels (e) and (f) show the biases relative.

We also provide maps of retrieved and reference TIWC at discrete altitude levels and zonal averages of retrieved and reference TIWC for different longitude bands in Fig. RC1.3 to Fig. RC1.7. While these results provide further evidence

[Figure]

Figure RC1.2: Zonally averaged distributions of retrieved TIWC and 2C-ICE-based reference TIWC for the CPCIR and GridSat test datasets over the domain covered by CCIC. Panels (a) and (b) show the retrieved and reference TIWC for the CPCIR observations aggregated to a resolution of 2.5°. Panel (c) show the truncated relative bias of the retrievals. Panels (d), (e), and (f) show the corresponding distributions of the GridSat-based retrieval.

of the ability of CCIC to capture the three-dimensional distribution of TIWC in the atmosphere, we do not plan to include them in the manuscript as it already contains a large number of figures.

[Figure]

Figure RC1.3: Distribution of retrieved and 2C-ICE-based reference TIWC measurements for CCIC CPCIR retrievals at three atmospheric levels. Column 1 displays the mean retrieved TIWC. Column 2 displays the corresponding reference TIWC. Column 3 displays the truncated relatived error. Rows contain the results for the atmospheric levels at altitudes of 5.5, 10.5 and 15.5 km, respectively.

**Changes in manuscript**

1. We will add the figure shown in Fig. RC1.1 together with the following paragraph discussing the results to the manuscript.

   **Changes starting in line 324:**

   The spatial distributions of mean retrieved and reference IWP concentrations from the test dataset are shown in Fig. 8. The distributions of the retrieved TIWP agree well with the distribution of the 2C-ICE measurements. Due to the low number of available 2C-ICE measurements in each 0.5 degree box, the relative bias field is noisy. The only region where the retrievals exhibit noticable relative biases is the southeast Pacific dry zone. This is likely caused by the low amount of ice clouds in this region in combination with increased relative retrieval uncertainties for low- and mixed-phase clouds. Overall, for 90% of all assessed 5° boxes the biases remain within $\pm 26.54.\%(\pm 55.78\%)$ for the CPCIR-based (GridSat-based) retrievals.

2. We will add the figure shown in Fig. RC1.2 together with the following paragraph discussing the results to the manuscript.

[Figure]

Figure RC1.4: As Fig. RC1.3 but for GridSat-based CCIC retrievals.

[Figure]

Figure RC1.5: As Fig. RC1.2 but for longitudes within $-180°$ and $-160°$.

Longitudes: $[-40°, -20°]$

[Figure]

Figure RC1.6: As Fig. RC1.2 but for longitudes within $-40°$ and $-20°$.

Longitudes: $[20°, 40°]$

[Figure]

Figure RC1.7: As Fig. RC1.2 but for longitudes within $20°$ and $40°$.

**Changes starting in line 334:**

 Zonal means of all retrieved and reference TIWC estimates are displayed in Fig. 10. Although both retrievals exhibit a tendency to underestimate the TIWC at cloud top and overestimate it at cloud base, the spatial distribution of TIWC is represented well. In particular, the  retrievals correctly represent the double-peak structure caused by the seasonal variability of the ITCZ as well as the asymmetry of the TIWC distribution in the ITCZ and the storm tracks.

**Specific comments**

**Comment 1**

The title is not representative of the context of the manuscript. There is no mention of constructing a climatology or anything of that sort.

**Author response**

We thank the reviewer for pointing out that the introduction failed to properly present the CCIC project and the scope of the manuscript. We have reformulated the abstract to state that CCIC is a novel cloud-property dataset. Moreover, we have reformulated the two paragraphs starting in l. 65 of the revised manuscript to clearly state that the presented manuscript describes the first step towards the production of an ice water path climate record using the presented retrieval.

**Changes in manuscript**

1. We will make the following modifications to the abstract.

   **Changes starting in line 9:**

   The Chalmers Cloud Ice Climatology (CCIC)  is a novel cloud-property dataset that aims to provide an improved climate record of ice hydrometeor concentrations by applying state-of-the-art machine-learning techniques to retrieve ice cloud properties from globally gridded, single-channel geostationary observations that are readily available from 1980. CCIC  offers a novel perspective on the record of geostationary IR observations by providing spatially and temporally continuous retrievals of the vertically-integrated and vertically-resolved concentrations of frozen hydrometeors, typically referred to as ice water path (IWP) and ice water content (IWC). In addition to that, CCIC provides 2D and 3D cloud masks and a 3D cloud classification.

2. We will modify the following paragraph in the introduction.

   **Changes starting in line 84:**

    In this first article, we present the neural-network-based retrieval algorithm underpinning CCIC and validate it against independent measurements of  hydrometeor concentrations. This work constitutes the first step towards producing an updated climate record of TIWP estimates, which we plan to follow-up with the production and publication of TIWP estimates for the full record of available geostationary IR observations.

**Comment 2**

Line 41: "For the study of processes on annual and decadal scales it is therefore necessary to find ways to make better use of observations with a long record of availability". The authors should mention that several IWP records exist with annual and even decadal scales, such as the ones from MODIS, Aura MLS, Odin SMR, CloudSat, etc. As currently written, the introduction implies that such records do not exist.

**Author response**

We thank the reviewer for pointing out this shortcoming of our manuscript. We will extend the introduction with a list of currently available records of comparable TIWP estimates and their respective shortcomings.

**Changes in manuscript**

1. We will modify the introduction as follows.

   > **Changes starting in line 58:**
   >
   > Of the currently available cloud datasets providing global estimates of the ice water path, those derived from combined radar-lidar observations from the CloudSat and CALIOP satellite have to be considered the most accurate due to their ability to resolve the ~~diurnal cycle of clouds. On the other hand, datasets derived from passive microwave sensors, only capture precipitating ice particles and were found to be at low end of the spectrum of global TIWP estimates. Although being derived from geostationary sensors, and thus capable of resolving the diurnal cloud cycle, TIWP estimates from the ISCCP dataset were found to be very low and not agree well with spatial distribution inferred from CloudSat measurements (Eliasson et al., 2011)~~ vertical structure of clouds and the combination of active measurements at microwave, IR and visible wavelengths. However, due to the thin swath of these observations, their revisit time is of the order of a few weeks. Moreover, due to a technical failure, observations are limited to day-time measurements since April 2011, and day-and-night-time measurements are only available between 2006 and 2011. Although global estimates of water paths are also provided by the MODIS, ISCCP-H series (Young et al., 2018), and PATMOS-x (Foster et al., 2023) products, which can be used to estimate the ice water path using provided cloud-phase information, these estimates are all limited to day-time observations. Furthermore, we were not able to find validation results for the ice water path estimates provided by the MODIS, ISCCP, and PATMOS-x datasets thus making it difficult for users to gauge their accuracy.

**Comment 3**

Further since CCIC provides IWC, the authors could compare partial IWP versus those records matching their respective altitude coverage. The comparison versus the campaigns is limited to a few periods and it is limited geographically.

**Author response**

As per the reviewer's suggestion, we will extended the analysis of the CCIC retrieval results to provide a more detailed assessment of the three-dimensional distribution or retrieved TIWC, as detailed in the response to major comment 1. The analysis is based on a full year of collocations with 2C-ICE estimates and thus covers the full geographical extent of the CCIC retrievals.

While the comment seems to suggest to also extend our comparison to estimates from limb-sounding instruments such as Aura MLS or Odin SMR, we choose not to do this as we do not think that this would offer any benefit over the comparison against the CloudSat/CALIPSO-based estimates.

Finally, we would like to point out that, while the individual field campaigns used in the validation are naturally limited in their spatial and temporal coverage, they cover both tropical and mid-latitude climate regimes and extend from instantaneous to seasonal time scales. With this, they exceed the scope of most validation efforts of TIWP/TIWC products that were able to find in published literature (Deng et al., 2010, 2013; Eriksson et al., 2008; Wu et al., 2008; Barker et al., 2008).

**Changes in manuscript**

See response to major comment 1.

**Comment 4**

Line 17: "considerable skill" is a qualitative description please provide a more quantitative description.

**Author response**

We will reformulate the sentence in question to provide quantitative accuracy estimate derived from independent test data.

**Changes in manuscript**

> **Changes starting in line 16:**
>
> A fully convolutional quantile regression neural network constitutes the core of the CCIC retrieval, providing probabilistic estimates of IWP and IWC. The network is trained against CloudSat retrievals using 3.5 years of global collocations.  Assessed on a held-out test  dataset, the CCIC-provided IWP and IWC estimates achieve correlations exceeding 0.7 and 0.6, respectively, and biases better than $-5\,\%$ and $-2\,\%$ demonstrating considerable skill in  estimating both IWP and IWC.

**Comment 5**

Line 20: "first order" is a qualitative description please provide a more quantitative description.

**Author response**

We will remove the formulation 'first order' and instead list the linear correlation coefficient and bias of the TIWC estimates in the previous paragraph.

**Changes in manuscipt**

See changes in reponse to specific comment 4.

**Comment 6**

Line 45: please describe the rationale behind only using the 11micron channel. Presumably additional channels could provide more information.

**Author response**

We chose the 11 μm channel because it provides the best temporal and spatial coverage throughout the available record of geostationary satellite observations. While the GridSat B1 product also includes visible and water vapor imagery, which could likely help to improve the retrieval, they are not always available and therefore not considered in the current CCIC retrievals. We will revise the paragraph in question to include the motivation for this design choice.

**Changes in manuscript**

> **Changes starting in line 75:**
>
> CCIC  provides estimates of TIWP and several other cloud properties from a single IR window channel  centered around 11 μm. Although these observations primarily provide information on the temperature of the atmosphere at the cloud top,  the 11 μm  channel provides the best availability among currently available gridded geostationary observation datasets (Knapp et al., 2011) and thus allows producing a long time series of spatially and temporally continuous TIWP measurements albeit limited to latitudes within $-60°N$ to $60°N$.

**Comment 7**

Line 55: "Estimates of TIWP differ widely between". Please give the ranges, this would allow you to later show how well (or bad) CCIC estimates are.

**Author response**

We will rewrite the relevant parts of the introduction and include the ranges of disagreement between satellite-based IWP estimates.

**Changes in manuscript**

**Changes starting in line 52:**

 estimates of the  ** **

~~TIWP shall be understood here as the vertically-integrated amount of all types of frozen hydrometeors. In currently available ice water path retrievals based on passive observations it is not always clearly defined which type of hydrometeorsare considered. As a consequence, TIWP is not very well constrained by currently available observations and there remain large differences in ice hydrometeor concentrations between different models (Waliser et al., 2009; Eliasson et al., 2011; Duncan and Eriksson, 2018). Estimates of TIWP differ widelyobservational datasets(Duncan and Eriksson, 2018)While this is, at least partly, due to the inherent limitations of different observing techniques and the significant impact of uncertain microphysical assumptions on TIWP estimates, additional factors limit the potential of currently availableto inform studies of cloud processes. For example, datasets derived from sensors in sun-synchronous orbits such as the CloudSat CPR, MODIS or AVHRR are typically not able~~. The principal reasons for these discrepancies are differences in the sensitivity of the underlying sensors to ice particles of different sizes and uncertain assumptions on the microphysical properties of the observed clouds. Moreover, it is not always clearly defined whether the estimates provided by a product include all frozen hydrometeors or are limited to either only suspended or precipitating particles.

**Comment 8**

Line 87: why not use lat lon info as well? And day of the year?

**Author response**

We do not include spatial and/or seasonal context in the retrieval input data as we want the underlying neural network to learn relations between the satellite observations and the corresponding cloud properties and not their variability with respect to geographical location and season. Since the training data period is limited from mid-2006 through 2009, including geographical coordinates and seasonal information could limit the retrieval's ability to reproduce changes in the regional or seasonal variability of clouds outside of the training data period.

**Comment 9**

Line 108: The use of "2D" here is confusing since the authors are talking about profiles, I suggest deleting it.

**Author response**

We will replace '2D' with 'horizontal'.

**Changes in manuscript**

> **Changes starting in line 128:**
>
> TIWP, TIWC, a  horizontal cloud mask indicating the presence of a cloud anywhere in the vertical profile, and  vertically-resolved cloud classification following the 2B-CLDLASS product.

**Comment 10**

Line 113 – Line 119: A schematic of this entire procedure will be appreciated. Also, what is the treatment for the uncertainties in 2C-ICE

**Author response**

We acknowledge the importance of making our training-data-generation process transparent and reproducible. To this end, we have published all relevant code in the repository accompanying this manuscript. However, since the manuscript already contains a large number of figures, we chose not to include a schematic of the data extraction process as we consider it of minor interest for the general audience.

We will add a sentence referring the interested reader to the relevant code to the revised manuscript.

Regarding the uncertainties of the 2C-ICE product: We treat the 2C-ICE estimates as ground-truth and do not make any effort to model the associated uncertainties. Synetergistic radar-lidar retrievals of ice hydrometeor concentrations have to be regarded as the most most accurate global measurements of TIWC and TIWP. Although even these combined radar-lidar estimates remain affected by significant systematic uncertainties, largely due to the underlying microphysical assumptions on particle shape and distribution, these uncertainties are not well characterized due to the limited amount of work that compares them with in-situ measurements (Deng et al. (2010, 2013) are the only studies that we are aware of). Since what ultimately matters to future users of CCIC is the uncertainty in the estimates provided by CCIC, we chose to extensively validate the resulting CCIC estimates instead of trying to handle uncertainties in the 2C-ICE product upfront.

**Changes in manuscript**

> **Changes starting in line 169:**
>
> Table 3 shows the sizes of the training, validation and test datasets. All code required to generate the training datasets are made available through the code repository accompanying this article (Amell and Pfreundschuh, 2023).

**Comment 12**

Line 136: "Training scenes of 384×384 pixels". Is the geographical size of this scene important? Is there an impact for using smaller or bigger scenes? Why this particular size.

**Author response**

We chose this size since it allows use to extract randomly rotated crops of size 256×256 pixels without generating invalid values. The scene size of 256×256, which is ultimately used in the retrieval, was chosen because it corresponds to scenes covering more than 900 km in zonal and meridional extent. The resulting scenes should thus contain information on the mesoscale and, to limited extent, synoptic-scale context of the retrieval.

To make this point clear, we have reformulated the paragraph in question to provide a better description of the training-dataset generation and the underlying motivation.

**Changes in manuscript**

**Changes starting in line 155:**

 The collocated geostationary input observations and CloudSat-Calipso-based reference data are used to generate the training dataset, which consists of scenes of a horizontal extent of $384 \times 384$  input-observation pixels. The scene size of 384 pixel was chosen as it allows for the extraction of randomly-rotated center crops of size $256 \times 256$ pixels, which is ultimately for training and inference. The extent of 256×256 pixels was chosen as it results in scenes exceeding 900 km in zonal and meriodional extent and thus should contain information on the mesoscale and, to some extent, also the synoptic-scale context of the retrieval.

**1.1 Comment 13**

Line 136: "The process involved randomly selecting a pixel with valid reference data as the starting point and then adding a random zonal offset." I don't really understand this please clarify

**Author response**

We have reformulated the paragraph in question to describe the training-data generation process more clearly.

**Changes in manuscript**

**Changes starting in line 160:**

Scenes are extracted by randomly selecting a pixel with valid reference data as the  center point for the scene. Then, a random zonal shift of up to 50 pixels east or west is applied to the scene so that the relative position of the CloudSat swath within the scene is randomized. This process was repeated until all pixels with valid reference data were included in at least one training scene. Scenes with less than 20% of valid  reference data pixels were discarded.

**Comment 14**

Line 140: lower → coarser

**Author response**

We will adopt this suggestion in the revised version of the manuscript.

**Changes in manuscript**

**Changes starting in line 165:**

Figure 1 shows that there is a clear difference in the spatial distributions of the collocations between the two IR data products, which is a result of the fixed overpass times of CloudSat and the  coarser temporal resolution of the GridSat data.

**Comment 15**

Figure 2 caption: Why is the brightness temperature normalized?

**Author response**

Normalizing the inputs to neural networks is common practice and generally leads to better results and faster training convergence.

**Comment 16**

Line 155: Good is the enemy of great. The authors should explore tuning of these parameters or rephrase this sentence to state that minimal tuning was required.

**Author response**

It is generally acknowledged that exhaustive tuning of all architecture-related hyperparameters of a neural network model is too resource consuming to be practically feasible and principled architecture search remains and activate area of machine-learning research (Ren et al., 2021). We therefore consider the reviewer's request out of scope for our work.

[Figure]

Figure RC1.8: Spatial coverage of  TWC   during the different HAIC-HIWC campaigns and collocated with CCIC retrievals. Flights during January and February 2014 were performed over Darwin, Australia and the surrounding oceans. Flights during May 2015 were performed out of Cayenne, French Guyana. Flights during August 2015 were performed out of Fort Lauderdale, USA. Flights in August 2018 were based out of Fort Lauderdale, Palmdale on the west coast of the USA and Kona on Hawaii. The map background is based on NASA Visible Earth imagery.

**Comment 14**

Figure 3, and 4 captions: specify the locations. for example, Aug 2015, and Jul-Aug 2018 flights were over the US and the US nearest oceans. Or, Flights took place over the Olympic Peninsula in the Pacific Northwest of the United States.

**Author response**

We will add the requested flight locations to the captions of Fig. 3 and 4.

**Changes in manuscript**

We have updated the figure captions of Fig. 3 and Fig.4. The figures and updated captions are shown in Fig. RC1.8 and Fig. RC1.9.

[Figure]

Figure RC1.9: Flight paths of the UND Citation and NASA ER-2 aircraft during the OLYMPEX campaign over the Olympic Peninsula in the pacific North-West region of the USA. The map background is based on NASA Visible Earth imagery.

**Comment 15**

Line 229: Which of the four periods is the Darwin campaign?

**Author response**

We will extend the sentence in question to state that the Darwin campaign is the first set of flights of the HAIC-HIWC campaign.

**Changes in manuscript**

> **Changes starting in line 255:**
>
> In addition to the in-situ measurements collected during all flights of the HAIC-HIWC campaigns, the  first campaign in Darwin, Australia also included 95-GHz cloud radar measurements from the Radar Airborne System Tool for Atmosphere (RASTA) radar flown onboard the Falcon 20 of the Service des Avions Francais Instrumentations pour la Recherche en Environnement (SAFIRE) that are publicly available.

**Comment 16**

**Author response**

Following the reviewer's suggestion, we have replaced 'W-band' by '94-GHz'.

**Changes in manuscript**

> **Changes starting in line 263:**
>
> As part of the campaign,  94-GHz cloud radar observations were performed by the NASA Cloud Radar System (CRS, Li et al., 2004) on board the NASA ER-2 aircraft.

**Comment 17**

Line 247: Which year?

**1.1.1 Author response**

The year was 2019. We will include this information in the revised manuscript.

**1.1.2 Changes in manuscript**

> **Changes starting in line 274:**
>
> For this study we use one year (2019) of radar measurements (Delanoë and Haeffelin, 2023) from the 95-GHz Bistatic Radar System for Atmospheric Studies (BASTA, Delanoë et al., 2016) from the site in Palaiseau, France.

**Comment 18**

Figure 5: panels g and h should say Cloud classification and cloud classification (retrieved) respectively.

**Author response**

We will update the panel titles in the revised manuscript to reflect this information.

**Changes in manuscript**

The updated figure is shown in Fig. RC1.10

[Figure]

Figure RC1.10: Retrieved and reference cloud properties from a CloudSat overpass over a mid-latitude cyclone over the North-American West Coast on 3 January 2019. Panel (a) shows the CPCIR input observations. Panel (b) displays a map of the retrieved TIWP over the region as well as the dominant cloud type, which is defined as the most frequent non-clear cloud class (as defined in panels (g) and (h)) in the atmospheric column. Panel (c) shows the retrieved and reference TIWP along the CloudSat ground track marked by the blue line in Panel (a). Panel (d) shows the TIWC from 2C-ICE along the CloudSat ground track. Panel (e) shows the corresponding retrieved TIWC. Panel (f) shows the retrieved 3D cloud mask. Panel (g) shows the cloud classification from the 2B-CLDCLASS product. Panel (h) shows the corresponding retrieved cloud classes.

**Comment 19**

Figure 6 caption should mention cloudsat somewhere, as well as the period use for this comparison.

**Author response**

We will add the requested information to the figure caption.

[Figure]

Figure RC1.11: Conditional distributions of retrieved TIWP conditioned on  2C-ICE-based reference TIWP  for the test samples from the year 2010. Panel (a) shows the distribution for the CPCIR input observations;  panel (b) shows the corresponding distributions for the GridSat dataset. The displayed bias and correlation coefficients are computed using all test samples including those outside the range of the scatter plot.

**Changes in manuscript**

The figure with the updated caption is shown in Fig. RC1.11

**Comment 20**

Section 3.2.1. It is not clear which period this comparison cover.

**Author response**

To make it clear to the reader that all results in this sub-section are derived using the independent test dataset, we will add an introductory sentence to Section 3.2. Furthermore, we will include this information in the all figure captions from this section.

**Changes in manusript**

1. We will add the following introductory sentence to the beginning of Sect. 3.2.

   **Changes starting in line 310:**

   The following sections quantitatively assess the accuracy of the CCIC retrieval on the independent test dataset, which consists of all CloudSat collocations from the year 2010.

2. We have updated the captions of all figures in this section to clearly state that the results were derived from the test dataset.

**Comment 21**

Line 297: This should be shown as a separate subsection to emphasize its importacnce: Zero order comparison of IWP and IWC (for example)

**Author response**

Following the reviewers suggestion, we have added a new section titled 'Consistency of retrieved TIWP and TIWC profiles' and moved the discussion of the consistency of the retrieved TIWP and TIWC there.

**Changes in manuscript**

> **Changes starting in line 339:**
>
> Since TIWC is retrieved on evenly-spaced altitude levels, the differences in the relative retrieval biases between TIWP (Fig. 6) and TIWC (Fig. 9) indicate small, systematic differences between the retrieved TIWP  and the column-integrated retrieved TIWC. When the retrieved TIWP and the column-integrated retrieved TIWC are compared directly, their linear correlation is 1.0 and the overall bias  at most 2.52%. Compared to the reference TIWP, the integrated retrieved TIWC yields slightly smaller biases (−0.72% and −1.90% for CPCIR and GridSat, respectively) but similar correlation values. However, since these differences are of the order of a few percent, they can be considered negligible compared uncertainties in the reference data.

**Comment 22**

I think the whole classification is barely working and the authors should just not show any of those results.

**Author response**

We respectfully disagree with the reviewer's comment but acknowledge that results presented in the first version of the manuscript may give an overly negative view of the retrieval's capabilities. We would like to point out, however, that the classification results from the case study presented in Fig. 5 demonstrate the ability of the retrieval to distinguish the types of the principal cloud systems in the scene. To provide a more comprehensive picture of the retrieval's classification skill, we will add curtain plots (conceptually similar to the ones the reviewer requested to demonstrate the skill of TIWC retrieval) showing the spatial distributions of the retrieved and reference cloud classes with respect to latitude and altitude. The results, shown in Fig. RC1.12 below, show good agreement between the distributions of the retrieved and reference cloud classes for all cloud classes except the stratus (St) class, whose frequency in the training dataset is only 0.03 %. We consider this compelling evidence that the CCIC retrieval, in fact, has skill in classifying different cloud types.

In addition to adding the curtain plots to the manuscript, we will add text that discusses the results and points out that the confusion matrix shown in Fig. 10 assesses the classification of cloud layers at a vertical resolution of 1 km, which leads to high uncertainties in the classification of individual layers.

**Changes in manuscript**

1. We will add the figure shown in Fig. RC1.12 to the manuscript.

2. We will add the following paragraph extending the discussion of the classficiation results.

> **Changes starting in line 372:**
>
> While the confusion matrix shown in Fig. 12 suggests high uncertainties for the classification of the 1-km vertical levels used by CCIC, the results from the case study shown in Fig. RC1.10 indicate that the retrieval can nonetheless successfully identify the dominant cloud systems in the scene and their vertical extent. To assess the ability of the retrieval to distinguish different cloud systems on larger scales, Fig. RC1.12 shows the frequency of occurrence of the different retrieved and reference cloud types by altitude and latitude band. As these results show, the spatial distribution of the cloud types agrees well with the reference distributions for all cloud types except St, which is never detected. These results confirm that the retrieval has certain skill in distinguishing different cloud systems and their vertical extent despite uncertainties in the classification of individual layers.

**Comment 23**

Figure 13 is missing the conditional mean line

**Author response**

We have made the conscious decision to not show conditional mean lines for the validation results as some of the campaigns have very few samples causing the conditional mean lines to become very noisy. While the HAIC-HIWC in-situ data used in Fig. 13 contains sufficient samples to show the conditional mean lines (see Fig. RC1.13 below) we feel that we would have to add conditional mean lines to all following scatter plots to be consistent. Since we do not think that the conditional mean lines add significant information to the scatter plots, we have decided against showing conditional mean lines for the validation results.

**Comment 24**

Line 417: CCIC is not representing the diurnal variability well, it is really flat.

[Figure]

Figure RC1.12: Spatial distribution of retrieved cloud classes and 2B-CLDCLASS-based reference cloud classes for the test samples from the year 2010. Each row of panels shows the distribution of one of the 8 cloud classes distinguished by the 2B-CLDCLASS product. The first column shows the results retrieved from CPCIR observations while the second column shows the corresponding reference distribution. Column three and four show the corresponding results for retrievals based on GridSat observations.

[Figure]

Figure RC1.13: As in Fig. 13 from the preprint, but with conditional mean lines.

**Author response**

While the CCIC results certainly do not represent the diurnal variability perfectly, the retrieved and reference diurnal cycles show a high degree of correlation. To make this point clear we will add a table containing the relative biases and linear correlation coefficients of the retrieved mean TIWP to the manuscript. These results show that the correlation of the diurnal cycles calculated over the full year is is 0.86 (0.97) for CPCIR-based (GridSat-based) retrievals and does not fall below 0.74 for any of the assessed three-month periods.

**Changes in manuscript**

1. We will add the tabe shown in table RC1.13 to the manuscript.

2. We will reformulate the the discussion of the diurnal cycles.

> **Changes starting in line 459:**
>
> Figure 21 shows diurnal cycles of TIWP retrieved from the ground-based radar and the CCIC retrievals. Calculated over the full year 2019, the CCIC retrievals reproduce the TIWP peak in the morning but underestimate the reduction in TIWP occurring around 15 h. The CCIC retrievals capture most of the seasonal variation of the diurnal cycle. The exception  are the summer months, during which the CCIC results capture the general shape of the diurnal cycle but underestimate the magnitude of the variation. Nonetheless, as shown in table 5, the correlation of the retrieved diurnal cycles and those derived from radar simulations using the Large-Plate Aggregate particle exceed 0.7 during all seasons. Due to their lower temporal resolution, the GridSat-based results generally exhibit weaker variations than the CPCIR retrievals but yield higher correlations compared to the reference diurnal cycles calculated at 3h resolution.
>
> Overall, it is notable that the CCIC retrievals manage to reproduce the diurnal and seasonal variation relatively well, as presented in Table RC1.13: the linear correlation between retrievals from CCIC and the Cloudnet radar used is at least 0.74 for any of the assessed three-month periods. It is important to note that CloudSat measurements are essentially limited to two discrete local overpass times due to the sun-synchronous orbit of the satellite. Therefore, they cannot resolve the diurnal cycle of cloud properties. The good agreement with the ground-based measurements shows that, despite being based on CloudSat measurements, the CCIC retrieval can reproduce diurnal variations in TIWP. The CCIC retrievals thus have the potential to provide an important novel perspective on ice clouds in the atmosphere.

Table RC1.13: Relative bias and linear correlation coefficient of the diurnal cycles retrieved using CCIC compared to those derived from ground-based cloud-radar observations.

| Time period | CPCIR | | GridSat | |
|---|---|---|---|---|
| | Bias [%] | Correlation coeff. | Bias [%] | Correlation coeff. |
| All year | 27.43 | 0.86 | 34.59 | 0.97 |
| DJF | 37.97 | 0.92 | 23.58 | 0.97 |
| MAM | 53.04 | 0.81 | 73.47 | 0.92 |
| JJA | 45.76 | 0.74 | 30.92 | 0.96 |
| SON | -0.66 | 0.84 | 25.79 | 0.92 |

**Comment 25**

Line 438: provide an estimate of the uncertainties associated with the estimates of the ice hydrometeors.

**Author response**

We have extended the sentence in question to reference the validation study of A-train-based IWC retrievals by Deng et al. (2013) and mention the biases of up to 59 % compared to in-situ measurements.

**Changes in manuscript**

**Changes starting in line 487:**

The biases with respect to in-situ measurements were within or close to 50 % for both flight campaign series, which is an encouraging result considering the overall uncertainty associated with estimates of ice hydrometeorsthat biases in combined radar-lidar retrievals, which are taken as reference estimates here, can be up to 59% for comparisons against in-situ measurements (Deng et al., 2013).

**Comment 26**

Line 456: "Despite these encouraging results, CCIC should still be considered a proof of concept. CCIC's principal objective remains to explore the potential of modern deep-learning techniques to expand the observational climate record of ice clouds". This should be mentioned upfront in the abstract and the introduction.

**Author response**

Since we have in the mean time processed the full observational record of available geostationary observations, it is not adequate anymore to consider CCIC

Table RC1.13: Cloud class frequencies, in percent, from all levels in the training set and total fraction of cloudy profiles, defined as a profile with at least one cloudy level.

| Cloud class | GridSat | CPCIR |
|---|---|---|
| No cloud | 91.7 | 92.2 |
| Cirrus | 1.4 | 1.5 |
| Altostratus | 2.2 | 2.0 |
| Altocumulus | 0.6 | 0.6 |
| Stratus | 0.03 | 0.03 |
| Stratocumulus | 0.9 | 0.9 |
| Cumulus | 0.3 | 0.3 |
| Nimbostratus | 2.0 | 1.5 |
| Deep convection | 0.9 | 0.9 |
| Cloudy  profile* | 46.7 | 44.3 |

merely a proof of concept. We will therefore the sentence from the revised manuscript.

**Changes in manuscript**

> **Changes starting in line 507:**
>
>  CCIC's  objective is to improve the observational climate record of ice-hydrometeor concentrations using modern deep-learning techniques.

**1.2 Comment 27**

Table A1: "Cloudy pixel" Cloudsat or retrieved? how come the cloudy pixel is 40ish while the no cloud is 97 %, please clarify

**Author response**

"Cloudy pixel" refers to cloudy profiles containing a cloud anywhere at the 20 vertical levels used by CCIC. The 97 %, on the other, hand refer to the fraction of non-cloudy levels. To make this point clearer we will replace 'cloudy pixel' with 'cloudy profile' in Table 1 and update the caption.

**Changes in manuscript**

We will update the table and caption which are shown in table RC1.13.

**References**

A. Amell and S. Pfreundschuh. SEE-GEO/ccic: Pre-release for manuscript submission., Aug. 2023. URL `https://doi.org/10.5281/zenodo.8278127`.

H. W. Barker, A. V. Korolev, D. R. Hudak, J. W. Strapp, K. B. Straw-bridge, and M. Wolde. A comparison between cloudsat and aircraft data for a multilayer, mixed phase cloud system during the canadian cloudsat-calipso validation project. *Journal of Geophysical Research: Atmospheres*, 113(D8), 2008. doi: https://doi.org/10.1029/2008JD009971. URL `https://agupubs.onlinelibrary.wiley.com/doi/abs/10.1029/2008JD009971`.

J. Delanoë and M. Haeffelin. Radar data from palaiseau on 18 july 2023, 07 2023. URL `https://cloudnet.fmi.fi/file/ec16dad5-3047-4a29-a437-91fc6d55607e`. Data is volatile and may be updated in the future.

J. Delanoë, A. Protat, J.-P. Vinson, W. Brett, C. Caudoux, F. Bertrand, J. Parent du Châtelet, R. Hallali, L. Barthes, M. Haeffelin, and J.-C. Dupont. Basta: A 95-ghz fmcw doppler radar for cloud and fog studies. *Journal of Atmospheric and Oceanic Technology*, 33(5):1023 – 1038, 2016. doi: 10.1175/JTECH-D-15-0104.1.

M. Deng, G. G. Mace, Z. Wang, and H. Okamoto. Tropical composition, cloud and climate coupling experiment validation for cirrus cloud profiling retrieval using cloudsat radar and calipso lidar. *Journal of Geophysical Research: Atmospheres*, 115(D10), 2010. doi: 10.1029/2009JD013104.

M. Deng, G. G. Mace, Z. Wang, and R. P. Lawson. Evaluation of several a-train ice cloud retrieval products with in situ measurements collected during the sparticus campaign. *Journal of Applied Meteorology and Climatology*, 52(4):1014 – 1030, 2013. doi: 10.1175/JAMC-D-12-054.1. URL `https://journals.ametsoc.org/view/journals/apme/52/4/jamc-d-12-054.1.xml`.

D. I. Duncan and P. Eriksson. An update on global atmospheric ice estimates from satellite observations and reanalyses. *Atmospheric Chemistry and Physics*, 18(15):11205–11219, 2018. doi: 10.5194/acp-18-11205-2018. URL `https://acp.copernicus.org/articles/18/11205/2018/`.

S. Eliasson, S. A. Buehler, M. Milz, P. Eriksson, and V. O. John. Assessing observed and modelled spatial distributions of ice water path using satellite data. *Atmospheric Chemistry and Physics*, 11(1):375–391, 2011. doi: 10.5194/acp-11-375-2011. URL `https://acp.copernicus.org/articles/11/375/2011/`.

P. Eriksson, M. Ekström, B. Rydberg, D. L. Wu, R. T. Austin, and D. P. Murtagh. Comparison between early odin-smr, aura mls and cloudsat retrievals of cloud ice mass in the upper tropical troposphere. *Atmospheric Chemistry and Physics*, 8(7):1937–1948, 2008. doi: 10.5194/acp-8-1937-2008. URL `https://acp.copernicus.org/articles/8/1937/2008/`.

M. J. Foster, C. Phillips, A. K. Heidinger, E. E. Borbas, Y. Li, W. P. Menzel, A. Walther, and E. Weisz. Patmos-x version 6.0: 40 years of merged avhrr and hirs global cloud data. *Journal of Climate*, 36(4):1143 – 1160, 2023. doi: 10.1175/JCLI-D-22-0147.1. URL `https://journals.ametsoc.org/view/journals/clim/36/4/JCLI-D-22-0147.1.xml`.

K. R. Knapp, S. Ansari, C. L. Bain, M. A. Bourassa, M. J. Dickinson, C. Funk, C. N. Helms, C. C. Hennon, C. D. Holmes, G. J. Huffman, J. P. Kossin, H.-T. Lee, A. Loew, and G. Magnusdottir. Globally gridded satellite observations for climate studies. *Bulletin of the American Meteorological Society*, 92(7): 893 – 907, 2011. doi: 10.1175/2011BAMS3039.1.

L. Li, G. M. Heymsfield, P. E. Racette, L. Tian, and E. Zenker. A 94-ghz cloud radar system on a nasa high-altitude er-2 aircraft. *Journal of Atmospheric and Oceanic Technology*, 21(9):1378 – 1388, 2004. doi: 10.1175/1520-0426(2004)021¡1378:AGCRSO¿2.0.CO;2.

P. Ren, Y. Xiao, X. Chang, P.-y. Huang, Z. Li, X. Chen, and X. Wang. A comprehensive survey of neural architecture search: Challenges and solutions. *ACM Comput. Surv.*, 54(4), may 2021. ISSN 0360-0300. doi: 10.1145/3447582. URL `https://doi.org/10.1145/3447582`.

D. E. Waliser, J.-L. F. Li, C. P. Woods, R. T. Austin, J. Bacmeister, J. Chern, A. Del Genio, J. H. Jiang, Z. Kuang, H. Meng, P. Minnis, S. Platnick, W. B. Rossow, G. L. Stephens, S. Sun-Mack, W.-K. Tao, A. M. Tompkins, D. G. Vane, C. Walker, and D. Wu. Cloud ice: A climate model challenge with signs and expectations of progress. *Journal of Geophysical Research: Atmospheres*, 114(D8), 2009. doi: 10.1029/2008JD010015.

D. L. Wu, J. H. Jiang, W. G. Read, R. T. Austin, C. P. Davis, A. Lambert, G. L. Stephens, D. G. Vane, and J. W. Waters. Validation of the aura mls cloud ice water content measurements. *Journal of Geophysical Research: Atmospheres*, 113(D15), 2008. doi: https://doi.org/10.1029/2007JD008931. URL `https://agupubs.onlinelibrary.wiley.com/doi/abs/10.1029/2007JD008931`.

A. H. Young, K. R. Knapp, A. Inamdar, W. Hankins, and W. B. Rossow. The international satellite cloud climatology project h-series climate data record product. *Earth System Science Data*, 10(1):583–593, 2018. doi: 10.5194/essd-10-583-2018. URL `https://essd.copernicus.org/articles/10/583/2018/`.

**Response to RC2 on egusphere-2023-1953**

The Chalmers Cloud Ice Climatology: Retrieval implementation and validation
Preprint `https://doi.org/10.5194/egusphere-2023-1953`

Adrià Amell      Simon Pfreundschuh*      Patrick Eriksson

April 24, 2024

*Comments from the Anonymous Referee are presented in* gray *and our responses in black. Line numbers are given with respect to the revised manuscript.*

We thank the reviewer for investing their time into reading our manuscript and providing valuable feedback. We are convinced that the comments helped us to improve our manuscript. As the reviewer noticed, the stability of the retrieval will be assessed with a follow-up work. We address the specific comments below.

**Changes to the manuscript**

While the manuscript was in review we discovered a mistake in the radar retrievals from the Palaiseau cloud radar, which used a tropical instead of a mid-latitude PSD parametrization. For the revised manuscript, we have updated the results of the ground-based TIWP retrievals. This did not change the results considerably. We have also corrected a number of smaller mistakes in the figures included in the manuscript.

**1 Specific comments**

**Comment 1**

Since the input data is only tested on geostationary data with no input from polar-orbiting satellites, it is worth mentioning that high latitudes are not represented in the study/or something about the likely difficulties in retrieving IWP over snow-covered surfaces. This fact is pertinent since, as far as I understand, GridSat (or at least the new ISCCP-NG, another similar global geostationary dataset) may include polar-orbiting satellites to fill in the missing data at the poles in the future.

**Author response**

As suggested by the reviewer, we will add a remark regarding the limited latitudinal coverage of the CCIC retrievals to the paragraph starting in line 68 of the revised manuscript.
* * *
*simon.pfreundschuh@colostate.edu

Moreover we will add a paragraph to the discussion secion in which we discuss the prospects of applying the CCIC retrieval to high-latitude and polar regions.

**Changes in manuscript**

1. We will add the following remark to the introduction.

> **Changes starting in line 76:**
>
> Although these observations primarily provide information on the temperature of the atmosphere at the cloud top,  the $11\,\mu$m  channel provides the best availability among currently available gridded geostationary observation datasets [Knapp et al., 2011] and thus allows producing a long time series of spatially and temporally continuous TIWP measurements albeit limited to latitudes within $-60°N$ to $60°N$.

2. We will add the following paragraph to the discussion section.

> **Changes starting in line 538:**
>
> Since CCIC was designed to be applied to geostationary observations, its retrievals are currently limited to the range $-60°N$ ($-70°N$) to $60°N$ ($70°N$) for the CPCIR-based (GridSat-based) retrievals. We are confident that the approach could also be applied to high-latitude and polar regions using observations from polar-orbiting satellite such as those used by the PATMOS-x dataset. While cloud retrievals of low clouds over snow-covered surfaces present specific technical difficulties, the machine-learning-based approach could benefit from improved spectral information provided by AVHRR-type sensors and the increased coverage of CloudSat/CALIPSO observations at high latitudes.

**Comment 2**

The datasets apply inter-satellite normalization. " This is not obvious. One method of "normalizing" the geostationary satellites is to use spectra band adjustments to make all the satellite's 11-micron channels look like a particular sensor, for instance, the SEVIRI 11-micron channel. Is this how it was done? Either way, more information is needed here

**Author response**

While we do not apply any normalization ourselves, we used the GridSat and CPCIR data as is, which have been already normalized. We updated the text in question to provide a high-level summary of the normalization applied by these datasets.

**Changes in manuscript**

> **Changes starting in line 112:**
>
> Both datasets provide merged and gridded IR brightness temperatures from the channels closest to 11 $\mu$m from the global constellation of historical and current geostationary meteorological satellites.  Knapp et al. [2011] references therein detail that both datasets are provided after an intersallite normalization, i.e. viewing angle and parallax corrections, and GridSat, in addition,  employs a temporal calibration against High-Resolution Infrared Radiation Sounder (HIRS) near-11 $\mu$m channel data, targeting long historical analyses .

**Comment 3**

The existence of the 2C-ICE equivalent dataset, DARDAR, should be mentioned somewhere, at least in reference, and possibly half a sentence on why 2C-ICE was chosen as the reference dataset here.

**Author comment**

We will add this information to the revised manuscript.

**Changes in manuscript**

> **Changes starting in line 120:**
>
> The reference data for the CCIC retrieval targets is derived from two CloudSat products: the level 2 cloud scenario classification version R05 [2B-CLDCLASS, Sassen and Wang, 2008] and the level 2 CloudSat and CALIPSO ice cloud property version R05 [2C-ICE, Deng et al., 2010, 2013b, 2015]. The 2B-CLDCLASS product assigns each CloudSat radar bin one of nine different cloud classes (Table 1). This choice of reference data over similar products, e.g. DARDAR-cloud [Delanoë and Hogan, 2010], which can be regarded as the alternative to 2C-ICE, was motivated by the 2C-ICE product yielding smaller biases against in-situ measurements [Deng et al., 2013a]. We acknowledge that this study was performed using now-outdated versions of the retrievals, however, we were not able to find more recent validation studies involving the two products.

**Comment 4**

The authors rightfully point out that the largest source of uncertainties in IWP retrievals is the assumed ice particle microphysical model. However, nothing is mentioned about which microphysical model the 2C-ICE IWP retrievals assume. This needs to be mentioned, especially as it is rightfully considered when retrieving IWP from ground-based Radar.

**Author response**

We will add the information regarding the particle habit and PSD used by 2C-ICE to the discussion secion.

**Changes in manuscript**

> **Changes starting in line 515:**
>
> Since CCIC uses the 2C-ICE and 2B-CLDCLASS products as reference data, it will directly inherit their characteristics. In particular this means that CCIC is based on the same microphysical assumptions as these two products and will therefore reproduce their errors.  The 2C-ICE product uses a modified gamma particle size distribution (PSD) with a habit mixture of randomly oriented particles to retrieve TIWC from combined CloudSat and CALIPSO measurements [Deng et al., 2010], however, since does not provide detailed information regarding the properties of the particles it is not possible for us to assess the impact of those assumtpion . Nontheless, our validation showed that the resulting CCIC estimates agree reasonably well with in-situ measurements, thus instilling confidence in the reliability of both the reference data and the CCIC retrievals.

**Comment 5**

Cloudnet offers several years of W-band data and more sites than just the one, Palaiseau, in France. Why (only) this site? For instance, I don't know if it is too far North, but Norunda in Sweden would add sub-arctic conditions to the validation. A comment would suffice here

**Author response**

The Norunda radar at $60.0860°$, unfortunately, is right oustide the latitude covered by the CCIC CPCIR retrievals. We only looked into latitudes covered by both datasets. From these, the radar in Palaiseau can be considered to be the Cloudnet site with the most complete and high quality W-band radar data record, in particular for 2019, the year used (2023 is complete as well, but it is when the manuscript was written). Furthermore, we did not want to overload the paper with figures and restricted the validation to one radar. An additional reason for this is that running the radar-only retrievals for a full year computationally expensive. Hence the choice of Palaiseau.

We will add a sentence summarizing this motivation to Sect. 2.2.3.

**Changes in manuscript**

> **Changes starting in line 275:**
>
> For this study we use one year (2019) of radar measurements [Delanoë and Haeffelin, 2023] from the 95-GHz Bistatic Radar System for Atmospheric Studies (BASTA, Delanoë et al., 2016) from the site in Palaiseau, France. This Cloudnet site was chosen as it presented one of the most complete

W-band cloud radar data records, in particular for 2019, for the latitudes covered by the CCIC retrievals.

**References**

J. Delanoë and M. Haeffelin. Radar data from palaiseau on 18 july 2023, 07 2023. URL `https://cloudnet.fmi.fi/file/ec16dad5-3047-4a29-a437-91fc6d55607e`. Data is volatile and may be updated in the future.

J. Delanoë and R. J. Hogan. Combined cloudsat-calipso-modis retrievals of the properties of ice clouds. *Journal of Geophysical Research: Atmospheres*, 115 (D4), 2010. doi: 10.1029/2009JD012346.

J. Delanoë, A. Protat, J.-P. Vinson, W. Brett, C. Caudoux, F. Bertrand, J. Parent du Châtelet, R. Hallali, L. Barthes, M. Haeffelin, and J.-C. Dupont. Basta: A 95-ghz fmcw doppler radar for cloud and fog studies. *Journal of Atmospheric and Oceanic Technology*, 33(5):1023 – 1038, 2016. doi: 10.1175/JTECH-D-15-0104.1.

M. Deng, G. G. Mace, Z. Wang, and H. Okamoto. Tropical composition, cloud and climate coupling experiment validation for cirrus cloud profiling retrieval using cloudsat radar and calipso lidar. *Journal of Geophysical Research: Atmospheres*, 115(D10), 2010. doi: 10.1029/2009JD013104.

M. Deng, G. G. Mace, Z. Wang, and R. P. Lawson. Evaluation of several a-train ice cloud retrieval products with in situ measurements collected during the sparticus campaign. *Journal of Applied Meteorology and Climatology*, 52(4):1014 – 1030, 2013a. doi: 10.1175/JAMC-D-12-054.1. URL `https://journals.ametsoc.org/view/journals/apme/52/4/jamc-d-12-054.1.xml`.

M. Deng, G. G. Mace, Z. Wang, and R. P. Lawson. Evaluation of several a-train ice cloud retrieval products with in situ measurements collected during the sparticus campaign. *Journal of Applied Meteorology and Climatology*, 52(4): 1014 – 1030, 2013b. doi: 10.1175/JAMC-D-12-054.1.

M. Deng, G. G. Mace, Z. Wang, and E. Berry. Cloudsat 2c-ice product update with a new ze parameterization in lidar-only region. *Journal of Geophysical Research: Atmospheres*, 120(23):12,198–12,208, 2015. doi: 10.1002/2015JD023600.

K. R. Knapp, S. Ansari, C. L. Bain, M. A. Bourassa, M. J. Dickinson, C. Funk, C. N. Helms, C. C. Hennon, C. D. Holmes, G. J. Huffman, J. P. Kossin, H.-T. Lee, A. Loew, and G. Magnusdottir. Globally gridded satellite observations for climate studies. *Bulletin of the American Meteorological Society*, 92(7): 893 – 907, 2011. doi: 10.1175/2011BAMS3039.1.

K. Sassen and Z. Wang. Classifying clouds around the globe with the cloudsat radar: 1-year of results. *Geophysical Research Letters*, 35(4), 2008. doi: 10.1029/2007GL032591.